# Global catchment modelling using World-Wide HYPE (WWH), open data and stepwise parameter estimation

Berit Arheimer[1*], Rafael Pimentel[1,2], Kristina Isberg[1], Louise Crochemore[1], Jafet C.M. Andersson[1], Abdulghani Hasan[1,3], and Luis Pineda[1,4]

[1] *Swedish Meteorological and Hydrological Institute (SMHI), Folkborgsvägen 17, 60176 Norrköping, Sweden.*

[2] *University of Cordoba, Edf. Leonardo Da Vinci, Campus de Rabanales, 14071, Córdoba, Spain.*

[3] *Lund University Box 117, SE-221 00, Lund, Sweden.*

[4] *Yachay Tech University, Hacienda San José, Urcuquí, Ecuador.*

[*]Corresponding author: Berit Arheimer ( berit.arheimer@smhi.se )

## Abstract

Recent advancements in catchment hydrology (such as understanding catchment similarity, accessing new data sources, and refining methods for parameter constraints) make it possible to apply catchment models for ungauged basins over large domains. Here we present a cutting-edge case study applying catchment-modelling techniques with evaluation against river flow at the global scale for the first time. The modelling procedure was challenging but doable and even the first model version show better performance than traditional gridded global models of river flow. We used the open-source code of the HYPE model and applied it for >130 000 catchments (with an average resolution of 1000 km$^2$), delineated to cover the Earths landmass (except Antarctica). The catchments were characterized using 20 open databases on physiographical variables, to account for spatial and temporal variability of the global freshwater resources, based on exchange with the atmosphere (e.g. precipitation and evapotranspiration) and related budgets in all compartments of the land (e.g. soil, rivers, lakes, glaciers, and floodplains), including water stocks, residence times, and the pathways between various compartments. Global parameter values were estimated using a stepwise approach for groups of parameters regulating specific processes and catchment characteristics in representative gauged catchments. Daily and monthly time-series (> 10 years) from 5338 gauges of river flow across the globe were used for model evaluation (half for calibration and half for independent validation), resulting in a median monthly KGE of 0.4. However, the World-Wide HYPE (WWH) model shows large variation in model performance, both between geographical domains and between various flow signatures. The model performs best (KGE > 0.6) in Eastern USA, Europe, South-East Asia, and Japan, as well as in parts of Russia, Canada, and South America. The model shows overall good potential to capture flow signatures of monthly high flows, spatial variability of high flows, duration of low flows and constancy of daily flow. Nevertheless, there remains large potential for model improvements and we suggest both redoing the parameter estimation and reconsidering parts of the model structure for the next WWH version. This first model version clearly indicates challenges in large-scale modelling, usefulness of open data and current gaps in processes understanding. However, we also found that catchment modelling techniques can contribute to advance global hydrological predictions. Setting up a global catchment model has to be a long-term commitment as it demands many iterations; this paper shows a first version, which will be subjected

to continuous model refinements in the future. The WWH is currently shared with regional/local
modellers to appreciate local knowledge.

# 1. Introduction


Global hydrological models with various properties and structures are provided by several modelling
communities (see reviews by e.g. Bierkens et al., 2015 and Sood and Smakhtin, 2015), although it is
well recognized that uncertainties associated with existing models are high when simulating the
water cycle at the global scale (e.g. Wood et al., 2011). To overcome this, some communities suggest
hyper-resolution (Bierkens et al., 2015) while others propose better coupling with earth observations
(Sood and Smakhtin, 2015). In this paper, we argue to improve global hydrological-model
performance by applying methods from the catchment modelling community.
In catchment modelling the water balance and fluxes are calculated within water divides. The
geographic unit for process descriptions is thus a polygon defined by topography instead of a grid cell
defined by size, without physical boundaries. Recently, new topographic data with high resolution
(Yamazaki et al., 2017) enables definition of catchments globally. Having catchments as a calculation
unit makes it possible to apply an ecosystem approach and account for co-evolution of processes at
the landscape scale (e.g. Bloeschl et al., 2013). Model parameters can thus be linked to catchment
state from interacting entities and not only to aggregation of separated building blocks (grids) of the
catchment. The structure of the catchment model is usually a function of the modellers' hydrological
understanding and it is admitted that model parameters cannot be measured directly in many cases,
but have to be estimated (Wagener, 2003).
Catchment modellers' have a long tradition of evaluating model performance against observations of
river flow (e.g. Bergström and Forsman, 1973; Beven and Kirkby, 1979; Lindström et al., 1997) as this
is the integrated result of hydrological processes at the catchment scale and moreover, is relatively
easy to monitor. In the early 1970's, model parameters were calibrated using rather simple curve
fitting towards observed time-series of river flow in a specific catchment outlet (e.g. Bergström and
Forsman, 1973). Since then the methods for parameter estimation have become more sophisticated
with focus on uncertainties in parameter values. The catchment models themselves are normally
quick to run even on a personal computer, which has allowed the methods for evaluating and
calibrating catchment models to become computationally heavy, such as GLUE (Beven and Binley,
1992), DREAM (Laloy and Vrugt, 2012), or methods in the SAFE toolbox (Pianosi et al., 2015).
Nevertheless, with increasing computational capacity, these methods should be possible to apply
also across large domains with numerous river gauges.
The catchment community advocates the potential to advance science by addressing a larger domain
with multiple gauged catchments than just exploring one single catchment at a time (Falkenmark and
Chapman, 1989; Bloeschl et al., 2013; Hrachowitz et al., 2013; Gupta et al., 2014). One current trend
among catchment modellers' is thus to test their methods also at the continental scale (e.g.
Pechlivanidis and Arheimer, 2015; Abbaspour et al., 2015; Donnelly et al., 2016), where traditionally
other types of hydrological models were applied, using other modelling procedures and showing
other advantages than the methods used by the catchment modelling community (see e.g. Archfield
et al., 2015). Traditional global hydrological models are for instance water-balance and -allocation
models (e.g. Arnell, 1999; Vörösmarty et al., 2000; Döll et al., 2003; Mulligan, 2013) or
meteorological land-surface models (e.g. Liang et al., 1994; Woods et al., 1998; Pitman, 2003;
Lawrence et al., 2011) sometimes with more advanced routing schemes (e.g. Alferi et al., 2013). With
current evolution of catchment models, their performance can now be compared to more traditional
global and continental modelling approaches in the large-scale applications (Fig. 1).

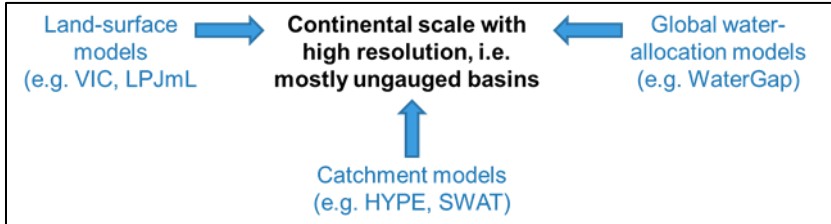

Figure 1. Different modelling communities who can now start comparing their results.

Bierkens et al., (2015) pose the question "how, if at all, it is possible to calibrate models at the global
scale". In fact, the catchment modelling community have developed several approaches to
regionalize parameter values for large domains, for instance by using: (i) the same parameters based
on geographic proximity (e.g. Merz and Blöschl, 2004; Oudin et al., 2008); (ii) regression models
between parameter values and catchment characteristics (Hundecha and Bárdossy, 2004; Samaniego
et al., 2010; Hundecha et al., 2016); (iii) simultaneous calibration in multiple representative
catchments with similar climatic and/or physiographic characteristics (e.g. Arheimer and Brandt,
1998; Fernandez et al., 2000; Parajka et al., 2007). Theoretically, these methods should be possible to
apply also on the global scale.
In this paper we test a variety of the latter method, using a stepwise approach (e.g. Strömqvist et al.,
2012; Pechlivanidis and Arheimer, 2015; Donnelly et al, 2016; Andersson et al., 2017) trying to isolate
hydrological processes and calibrate them separately against observed river flow in selected
representative basins across the entire globe (although some hydrological features such as large
lakes and floodplains were calibrated individually). This is an example of how to use the catchment
ecosystem approach assuming that hydrological processes are similar across the globe wherever the
catchments have evolved under similar conditions and have similar physiographic conditions.
The hypothesis tested in the present study states that it is now possible and timely to apply
catchment modelling techniques at the global scale, for which only gridded approaches have been
reported so far (Bierkens et al., 2015; Sood and Smakhtin, 2015). We address this hypothesis by
applying a catchment model world-wide and then evaluating the results, using statistical metrics for
streamflow time-series and signatures. To our knowledge, this is the first time a catchment model
was applied world-wide and evaluated against river flow across the globe. The catchments were
delineated and routed based on high-resolution topography (90 m) resulting in an average size
of ~1000 km$^2$ (WWH version 1.3). Our specific objective is to provide a harmonized way to predict
hydrological variables (especially river flow and the water balance) globally, and then the model set-
up can be shared for further regional refinement to assist in water management wherever
hydrological models are currently lacking. To address this objective, we (i) compile open global data
from >30 sources, including for instance topography and river routing, meteorological forcing,
physiographic land characteristics and in total some 20 000 time-series of river flow world-wide, (ii)
apply the open-source code of the Hydrological Predictions for the Environment, HYPE model
(Lindström et al., 2010), (iii) estimate model parameter values using a new stepwise calibration
technique addressing the major hydrological processes and features world-wide, and (iv) compute
metrics and flow signatures, and compare model performance with physiographic variables to judge
model usefulness. We then pose the scientific question: How far can we reach in predicting river flow
globally, using integrated catchment modelling, open global data and readily available time-series for
calibration?

## 2. The HYPE model


The development of the HYPE model was initiated in 2002, primary to support the implementation of
the EU Water Framework Directive in Sweden (Arheimer and Lindström, 2013). It was originally
designed to estimate water quality status, but is now also used operationally at the Swedish
hydrological warning service at SMHI for flood and drought forecasting (e.g. Pechlivanidis et al.,
2014). The water and nutrient model is applied nationally for Sweden (Strömqvist et al., 2012), the
Baltic Sea basin (Arheimer et al., 2012) and Europe (Donnelly et al., 2013). It also provides
operational hydrological forecasts for Europe at short-term and seasonal scale and it has been
subjected to several large-scale applications across the world, e.g. the Indian subcontinent
(Pechlivanidis and Arheimer, 2015) and the Niger River (Andersson et al., 2017). One of the main
drivers for HYPE applications has been climate-change impact assessments, for which its results have
been compared to other models in selected catchments across the globe (Geflan et al., 2017; Gosling
et al., 2017; Donnelly et al., 2017).
The HYPE model code (Lindström et al., 2010) represents a rather traditional integrated catchment
model, describing major water pathways and fluxes in a catchment ensuring that the mass of water is
conserved at each time step. Parameters are often linked to physiographic properties and the values
regulate the fluxes between water storages in the landscape and interaction with boundary condition
of the atmosphere, the oceans, and outlets of endorheic catchments, so called sinks (see section 4.1
and detailed model documentation at hypeweb.smhi.se). It is forced by precipitation and
temperature at daily or hourly time-step, and start by calculating the water balance of Hydrological
Response Units (HRUs), which is the finest calculation unit in each catchment. In the WWH set-up,
the HRUs were defined by land-cover, elevation and climate, without specific consideration to
further definition of soil properties. This was guided by recent studies indicating that soil water
storage and fluxes well related to vegetation type and climate conditions rather than soil properties
(e.g. Troch et al., 2009; Gao et al., 2014). HYPE has maximum three layers of soil and these were all
applied in the WWH, with a different hydrological response from each one for each HRU. The first
layer corresponds to some 25 cm, the second to some 1-2 meters and the third can be deep also
accounting for ground water. A specific routine can account for deep aquifers, but this was not
applied in the WWH due to lack of local or regional information of aquifer behaviour. HYPE has a
snow routine to account for snow storage and melt, while a glacier routine accounts for ice storage
and melt. Mass balances of glaciers were based on the observations provided in the Randolph Glacier
Inventory (Arendt et al., 2015) and fixed separately in the model set-up.
There are a number of algorithms available to calculate potential evapotranspiration (PET) in HYPE.
For the WWH we used the algorithms that had been judged most appropriate in previous HYPE
applications, giving Jensen-Haise (Jensen and Haise, 1963) in temperate areas, modified Hargreaves
(Hargreaves and Samani, 1982) in arid and equatorial areas, and Priestly Taylor (Priestly and Taylor,
1972) in polar and snow /ice dominated areas. River flow is routed from upstream catchments to
downstream along the river network, where lakes and reservoirs may dampen the flow according to
a rating curve. A specific routine is used for floodplains to allow the formation of temporary lakes,
which may be crucial especially in inland deltas (Andersson et al., 2017). Evaporation takes place
from all water surfaces, including snow and canopy. The HYPE source code, documentation and user
guidance are freely available at http://hypecode.smhi.se/.
# 3. Data

## 3.1 Physiographic data
For catchment delineation and routing, topographical data is needed, but none of the hydrologically
refined databases cover the entire land surface of Earth and therefore we had to merge several
sources of information (Table 1). Most of the globe (from 60S to 80N) is covered by GWD-LR (Global
Width Database of Large Rivers) 3 arc sec (Yamazaki et al. 2014 and 2017), apart from the very
northern part close to the Arctic Sea, for which HYDRO1K 30 arc sec (USGS) is used. For Greenland,
we used GIMP-DEM (Greenland Ice Mapping Project) 3 arc sec (Howat et al. 2014) and for Iceland the
National data from the meteorological office. For the latter we merged the catchments to better fit
the overall resolution, going from 27 000 catchments to 253. Each of the above datasets was used
independently in the delineation.
Additional data was gathered to help with defining catchments as the delineation of catchments can
be difficult in some environments. In flat areas we consulted previous mapping and hydrographical
information of floodplains, prairies and deserts (Table 1). Karstic areas are unpredictable due to lack
of subsurface information of underground channels crossing surface topography and thus needed to
be defined and evaluated separately. Finally, flood risk areas (UNEP/GRID-Europe ; Table 1) were
recognized as potentially important, enabling the use of model results in combination with hydraulic
models, and thus also had to be identified so that model results can be extracted for such
applications.

**Table 1.** Databases used for catchment delineation, routing and elevation in WWH version 1.3.

| Type | Dataset/Link | Provider/Reference |
| --- | --- | --- |

| Topography (Flow accumulation, flow direction, digital elevation, river width) | GWD-LR (3 arcsec) http://hydro.iis.u-tokyo.ac.jp/~yamadai/GWD-LR/ | Yamazaki et al., 2014; 2017; Howat et al., 2015 |
| | GIMP-DEM (3 arcsec) https://bpcrc.osu.edu/gdg/data/gimpdem | United State Geological Survey – (USGS) |
| | HYDRO1K (30 arcsec) https://lta.cr.usgs.gov/HYDRO1K | |
| | SRTM (3 arcsec) https://lta.cr.usgs.gov/SRTM | USGS |
| Non-contributing areas in Canada | Areas of Non-Contributing Drainage (AAFC Watersheds Project – 2013) https://open.canada.ca/data/dataset/67c8352d-d362-43dc-9255-21e2b0cf466c | Government Canada |
| Watershed delineation (Iceland) | IMO subbasins and main river basins http://en.vedur.is/hydrology/ | Icelandic Met Office (IMO) |
| Karst | World Map of Carbonate Rock Outcrops v3.0 http://digital.lib.usf.edu/SFS0055342/00001 | Ford (2006) |
| Global Flood Risk | Global estimated risk index for flood hazard http://ihp-wins.unesco.org/layers/geonode:fl1010irmt | UNEP/GRID-Europe |
| Floodplains | Global Lake and Wetland Database (GLWD) https://www.worldwildlife.org/publications/global-lakes-and-wetlands-database-lakes-and-wetlands-grid-level-3 | Lehner and Döll, 2004 |
| Desert areas | World Land-Based Polygon Features https://geo.nyu.edu/catalog/stanford-bh326sc0899 | University of New York |


For catchment characteristics governing the hydrological processes in HYPE, the ESA CCI Landcover
version 1.6.1 epoch 2010 (300 m) was the baseline for HRUs, but several other data sources were
used to adjust and add information to some hydrologically important features, such as glaciers, lakes,
reservoirs, irrigated crops, and climate zone (Table 2).

**Table 2.** Databases used to assign land cover, waterbodies and climate to catchments in WWH version 1.3.

| Type | Dataset/Link | Provider/References |
| --- | --- | --- |
| Land cover characteristics | ESA CCI Landcover v 1.6.1 epoch 2010 (300 m) https://www.esa-landcover-cci.org/?q=node/169 | ESA Climate Change Initiative - Land Cover project |
| Glaciers | Randolph Glacier Inventory (RGI) v 5.0 https://www.glims.org/RGI/randolph50.html | RGI Consortium |
| Greenland ice sheet | Greenland Glacier Inventory | Rastner et al, 2012 |
| Lakes | ESA CCI-LC Waterbodies 150 m  2000 v 4.0 https://www.esa-landcover-cci.org/?q=node/169 | ESA Climate Change Initiative - Land Cover project |
| Lakes | Global Lake and Wetland Database 1.1 (GLWD) https://www.worldwildlife.org/publications/global-lakes-and-wetlands-database-large-lake-polygons-level-1 | Lehner and Döll, 2004 |
| Lake depths | Global Lake Database  v2(GLDB) http://www.flake.igb-berlin.de/ep-data.shtml | Kourzeneva, 2010, Choulga, 2014 |
| Reservoirs and dams | Global Reservoir and Dam database v 1.1 (GRanD) http://www.gwsp.org/products/grand-database.html | Lehner et al., 2011 |

| Irrigation | GMIA v5.0 http://www.fao.org/nr/water/aquastat/irrigationmap/index10.stm MIRCA v1.1 http://www.uni-frankfurt.de/45218031/data_download | Siebert et al., 2013 Portmann et al., 2010 |
| --- | --- | --- |
| Climate classification | Köppen-Geiger Climate classification, 1976-2000, v June 2006 http://koeppen-geiger.vu-wien.ac.at/ | Kottek et al., 2006 |


## 3.2 Meteorological data

The WWH model uses time-series of daily precipitation and temperature to make calculations on a
daily time-step. All catchment models require initializations of the current state of the snow, soil and
lake (and sometimes river) storages. At the global scale, a seamless dataset for several decades is
necessary for consistent model forcing, to also cover hydrological features with large storage
volumes. For WWH version 1.3 precipitation and temperature were achieved from the Hydrological
Global Forcing Data (HydroGFD; Berg et al., 2018), which is an in-house product of SMHI that
combines different climatological data products across the globe. This global dataset spans a long
climatological period up to near-real-time and forecasts (from 1961 to 6 months ahead). The period
used in this study, is primarily based on the global (50 km grid) re-analysis product ERA-interim (Dee
et al., 2011) from ECMWF, which is further bias adjusted versus other products using observations,
e.g. versions of CRU (Harris and Jones, 2014) and GPCC (Schneider et al, 2014). The HydroGFD
dataset is produced using a method for bias adjustment, which is similar to the method by Weedon
et al. (2014) but additionally uses updated climatological observations, and, for the near-real-time,
interim products that apply similar methods. This means that it can run operationally in near-real-
time. The dataset is continuously upgraded and in the present study, we used the HydroGFD version
218  2.0.


## 3.3 Observed river flow

Catchment models need time-series of hydrological variables for parameter estimation and model
evaluation. Metadata and daily and monthly time-series from gauging stations were collected from
readily available open data sources globally (Table 3). In total, information from 21 704 gauging
stations could be assigned to a catchment outlet. Of these, time-series could be downloaded for 11
369 while 10 336 could only assist with metadata, such as upstream area, river name, elevation or
natural of regulated flow. The time-series were screened for missing values, inconsistency, skewness,
trends, inhomogeneity, and outliers (Crochemore et al., 2019). Stations representing the resolution
of the model ($\geq$1000 km$^2$) and with records of at least 10 consecutive years between 1981 and 2012
were considered for model evaluation. With these criteria, 5338 time-series were used for evaluating
overall model performance, of which 2863 represented independent model validation and 2475
were also involved in the stepwise model calibration (see section 4.2). In addition, 1181 stations not
fulfilling the criteria were added to increase the number of representative gauges to capture spatial
variability when estimating parameter values. In total, 6519 gauging stations were used for model
calibration and validation.

**Table 3.** Databases used for time-series of water discharge and location of gauging station when estimating
parameters and evaluating the model performance of WWH version 1.3.

| Data type | Short Name/Link | Coverage | Provider/References |
|---|---|---|---|
| Time-series + metadata | GRDC https://www.bafg.de/GRDC/EN/Home/homepage_node.html | Global | Global Runoff Data Center |
| " | EWA https://www.bafg.de/GRDC/EN/04_spcldtbss/42_EWA/ewa.html | Europe | GRDC – EURO-FRIEND-Water |
| " | Russian River data by Bodo, ds553.2 https://rda.ucar.edu/datasets/ds553.2/ | Former Soviet Union | Bodo, 2000 |
| " | R-ArcticNet v 4.0 http://www.r-arcticnet.sr.unh.edu/v4.0/index.html | Arctic region | Pan-Arctic Project Consortium |
| " | RIVDIS v 1.1 https://daac.ornl.gov/RIVDIS/guides/rivdis_guide.html | Global | Vörösmarty et al., 1998 |
| " | USGS https://waterdata.usgs.gov/nwis/sw | USA | U.S. Geological Survey |
| " | HYDAT https://www.canada.ca/en/environment-climate-change/services/water-overview/quantity/monitoring/survey/data-products-services/national-archive-hydat.html | Canada | Water Survey of Canada (WSC) |
| " | Chinese Hydrology Data Project https://depts.washington.edu/shuiwen/index.html | China | Henck et al., 2011 |
| " | Spanish Water Authorities https://www.mapama.gob.es/es/ministerio/funciones-estructura/organizacion-organismos/organismos-publicos/confederaciones-hidrograficas/default.aspx | Spain | Ecological Transition Ministry |
| " | WISKI https://vattenwebb.smhi.se/station/ | Sweden | Swedish Meteorological and Hydrological Institute |
| Metadata | CLARIS-project http://www.claris-eu.org/ | La Plata Basin | CLARIS LPB- project FP7 Grant agreement 212492 |
| " | CWC handbook http://cwc.gov.in/main/webpages/publications.html | India | Central Water commission (CWC) |
| " | SIEREM http://www.hydrosciences.fr/sierem/ | Africa | Boyer et al., 2006 |
| " | Regional data https://uia.org/s/or/en/1100058436 | Congo Basin | International Commission for Congo-Ubangui-Sangha Basin (CICOS) |
| " | National data http://www.bom.gov.au/water/hrs/ | Australia | BOM (Bureau of Meteorology) |
| " | Red Hidrometrica SNHN 2013 http://geo.gob.bo/geonetwork/srv/dut/catalog.search#/metadata/ff98cf17-f9a8-4a8d-b96c-bf623dd6b13b | Bolivia | Servicio Nacional de Hidrografía Naval |
| " | Estacoes Fluviometrica http://www.snirh.gov.br/hidroweb/ | Brazil | ANA (Agencia Nacional de Aguas) |
| " | Red Hidrometrica http://www.dga.cl/Paginas/default.aspx | Chile | DGA (Direccion General de Aguas) |
| " | Catalogo Nacional de Estaciones de Monitoreo Ambiental http://www.ideam.gov.co/geoportal | Colombia | IDEAM (Instituto de Hidrologia, Meteorologia y Estudios Ambientales) |

| | | | |
|---|---|---|---|
| " | Estaciones_Hidrologicas http://www.serviciometeorologico.gob.ec/geoinformacion-hidrometeorologica/ | Ecuador | INAMHI (Instituto Nacional de Meteorología e Hidrología) |
| " | National data http://www.senamhi.gob.pe/?p=0300 | Peru | SENAMHI (Servicio Nacional de Meteorologia e Hidologia del Peru) |
| " | National data http://www.inameh.gob.ve/web/ | Venezuela | IGVSB (Instituto Geográfico de Venezuela Simon Bolivar) |
| " | Conabio 2008 http://www.conabio.gob.mx/informacion/metadata/gis/esthidgw.xml?_httpcache=yes&_xsl=/db/metadata/xsl/fgdc_html.xsl&_indent=no | Mexico | Instituto Mexicano de Tecnología del Agua/CONABIO |
| " | Niger HYCOS http://nigerhycos.abn.ne/user-anon/htm/ | Niger river | World Hydrological Service System (WHYCOS) |
| " | National data https://www.dwa.gov.za/Hydrology/ | South Africa | Department Water & Sanitation, Republic of South Africa |
| " | National data http://publicutilities.govmu.org/English/Pages/Hydrology-Data-Book-2006---2010.aspx | Mauritius | Mauritius Ministry of Energy and Public Utilities |


## 4. Model setup


The WWH is developed incrementally, and the current version 1.3 was based on previous versions, where version 1.0 only included the most basic functions to run a HYPE model and was forced by MSWEP (Beck et al., 2017) and CRU (Harris and Jones, 2014). Version 1.2 included distributed geophysical and hydrographical features, and finally, version 1.3 (described below) included estimated parameter values and was forced by the meteorological dataset Hydro-GFD, which also provides operational forecasts at a 50 km grid (Berg et al., 2017). Gridded forcing data were linked to catchments using the grid point nearest to the catchment centroid. Dynamic catchment models need to be initialised to account for adequate storage volumes, which may, for instance, dampen or supply the river flow based on catchment memory (e.g. Iliopoulou et al., 2019). The WWH was initialized by running for a 15-year warm-up period 1965-1980, which was judged to be enough for more than 90% of the catchments by checking the time it takes for runs initialized 20 years apart to converge. Long initialization periods are needed for large lakes with small catchments, large glaciers, and sinks or rarely-contributing areas.

The current model runs at a Linux cluster (using nodes of 8 processors and 16 threads) with calculations in approximately 1 800 000 HRUs and 130 000 catchments covering the worlds land surface, except for Antarctica. The model runs in parallel in 32 hydrologically-independent geographical domains with a run time of about 3 hours for 30-year daily simulations. The methods applied for modelling and evaluation mostly follow common procedures used by the catchment modelling community, as described below.

260

## 4.1 Catchment delineation and characteristics

Catchment borders were delineated using the World Hydrological Input Set-up Tool (WHIST; https://hypeweb.smhi.se/model-water/hype-tools/), software developed at SMHI that is linked to the Geographic Information System (GIS) Arc-GIS from ESRI. By defining force-points for catchment outlets in the resulting topographic database (c.f. Table 1) and criteria for minimum and maximum ranges in catchment size, the tool delineates catchments and the link (routing) between them. By adding information from other types of databases, WHIST also aggregates data or uses the nearest grid for assigning characteristics to each catchment. WHIST handles both gridded data and polygons, and was used to link all data described in Section 2, such as land-cover, river width, precipitation, temperature, and elevation, to each delineated catchment. WHIST then compiles the input data files to a format that can be read by the HYPE source code. The software runs automatically, but also has a visual interface for manual corrections and adjustments. It may also adjust the position of the gauging stations to match the river network of a specific topographic database.

When setting up WWH, force-points for catchment delineation were defined according to:

- *Locations of gauging stations in the river network*: in total, catchments were defined for all 21 704 gauging stations which had an upstream area greater than 1000 km$^2$ (except for data sparse regions (500 – 1000 km$^2$). Their coordinates were corrected to fit with the river network of the topographic data, using WHIST and manually. Quality checks of catchment delineation were done towards station metadata and 88% of the estimated catchment areas were within +/-10% discrepancy towards metadata. These catchments were used in further analysis for parameter estimation or model evaluation; however, not all of these sites provided open access to time-series (see Section 2.3).

- *Outlets of large lakes/reservoirs*: New lake delineation was done to solve the spatial mismatch between data of the water bodies from various sources (c.f. Table 2). The centroid of the lakes included in GLWD and GRanD was used as initialization points for a Flood Fill algorithm, applied over the ESA CCI Water Bodies, followed by manual quality checks. The outlet location was defined using the maximum upstream area for each lake. In total, around 13 000 lakes and 2500 reservoirs > 10 km$^2$ were identified globally. The new dataset was tested against detailed lake information for Sweden, which represents one of the most lake-dense regions globally. Merging data from the two databases and adjusting to the topographic data used was judged more realistic for the global hydrological modelling than only using one dataset.

- *Large cities and cities with high flood risk*: The UNEP/GRID-Europe database (Table 1) was used to define flood-prone areas for which the model may be useful in the future. The criteria for assigning a force point was city areas of > 100 km$^2$ (regardless of the risks on the UNEP scale) or city areas of 10-100 km$^2$ with risk 3-5 and an upstream area > 1000 km$^2$. This was only considered if there was no gauging station within 10 km from the city. This gave another 2 439 forcing points to the global model.

302     •  *Catchment size*: the goal was to reach an average size of some 1000 km$^2$, for practical
303         (computational) and scientific reasons, reflecting uncertainty in input data. Criteria in WHIST
304         were set to reach maximum catchment size of 3000 km$^2$ in general and 500 km$^2$ in coastal
305         areas with < 1000 m elevation (to avoid crossing from one side to another of a narrow and
306         high island or peninsula). Post-processing was then done for the largest lakes, deserts, and
307         floodplains, following specific information on their character (see data sources in Table 2).

Using this approach, the land surface of the Earth (i.e. 135 million km$^2$ when excluding Antarctica)
was divided into 131 296 catchments with a mean size of 1020 km$^2$ (5$^{th}$ percentile: 64 km$^2$; 50$^{th}$
percentile: 770 km$^2$; 95$^{th}$ percentile: 2185 km$^2$). Flat land areas of deserts and floodplains ended up
with somewhat larger catchments, about 4500 km$^2$ and 3500 km$^2$, respectively. Around 23.8% of the
land surface did not drain to the sea but to sinks (Fig. 2), the largest single one being the Caspian Sea.
This water was evaporated from water surfaces but also percolated to groundwater reservoirs.
Moreover, several areas across the globe are of Karstic geology with wide underground channels,
which does not follow the land-surface topography. Sinks within Karst areas according to the World
Map of Carbonate Rock outcrops (Table 1) were linked to "best neighbour" and inserted to the river
network. The Canadian prairie also encompasses a large number of sinks due to climate and
topography, and there existed a national dataset from Canada with well-defined non-contributing
areas to adjust the routing in this area.

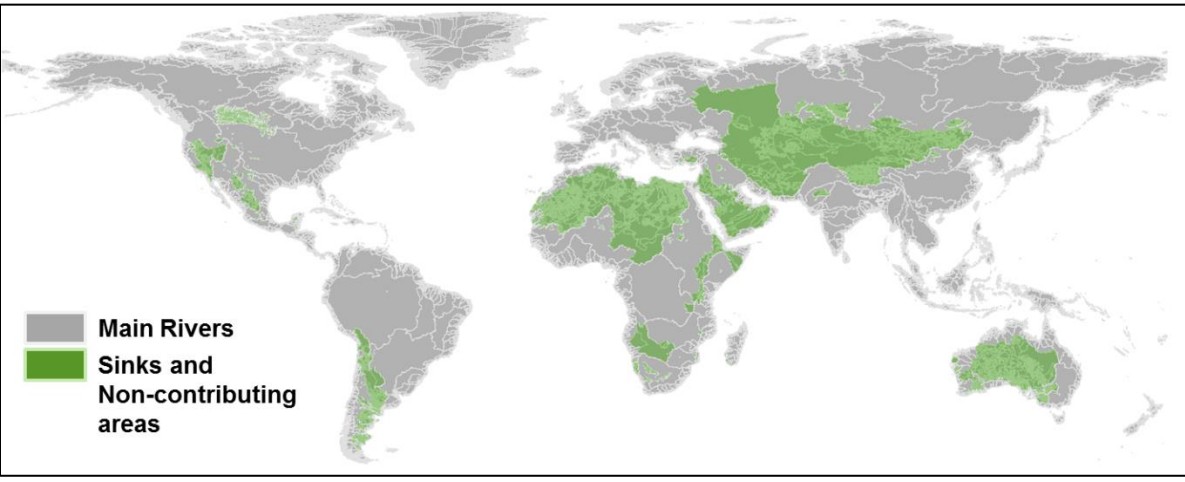

**Figure 2.** Major river basins and areas not contributing to river flow from land to the sea.

The land-cover data from ESA CCI LC v1.6 (Table 2) was used as the base-line for HRUs. It has 36
classes and subclasses and three of these were adjusted using additional data to improve the quality;
(i) by using glacier delineated by the RGI v5 and comparing spatially the outlines of both sources, we
avoided overestimation of the glacier area; (ii) by using GMIA and MIRCA in a data fusion algorithm
to create a more robust new irrigation database, we added irrigation information were is was missing
and underestimated; (iii) by combining several sources of water bodies (see Table 2) and spatial
analyses (e.g. a flood fill algorithm and geospatial tools) we differentiated one general class of
waterbodies into four: large lakes, small lakes, rivers, and coastal sea, which makes more sense in
catchment modelling. Five elevation zones were derived to differentiate land-cover classes with
altitude (0-500 m, 500-1000 m, 1000-2000 m, 2000-4000 m and 4000–8900 m) as the hydrological
response may be very different at different altitude due to vegetation growth and soil properties.
The land-cover at these elevations was thus treated as a specific HRU globally. In total, this resulted
in 169 HRUs.
All catchments were characterized according to Köppen-Geiger (Table 2) to assign a PET algorithm
(see section 3.2) but the characteristics did not include soil properties, which is common in
catchment hydrology. The approach when setting up HYPE was to use the possibility to assign
hydrologically active soil depth for the HRUs instead (see Section 2 on HYPE model), based on the
variability in vegetation, climate and elevation they represent as suggested by Troch et al. (2009) and
Gao et al. (2014). However, a few distinct soil properties were unavoidable beside the general soil to
describe the hydrological processes; these were impermeable conditions of urban and rock
environments, and infiltration under water and rice fields.

## 4.2 Stepwise parameter estimation
The method to assign parameter values for the global model domain aimed at finding (i) robust
values also valid for ungauged basins, as well as (ii) reliable process description of dominating flow
generation processes and water storage along the flow paths. The first aim was addressed by
simultaneous calibration in multiple representative catchments world-wide. Spatial heterogeneity
was accounted for by separate calibration of catchments representing different climate, elevation,
and land-cover globally. The second aim was addressed by applying a stepwise approach following
the HYPE process description along the flow paths, only calibrating a few parameters governing a
specific process at a time (Arheimer and Lindström, 2013). The estimated parameter values were
then applied wherever relevant in the whole geographical domain, i.e. world-wide. We estimated
parameters for 11 hydrological processes separately, where each process description includes
between 2 and 20 parameters (Table A1 in the Appendix). Some processes were calibrated for
specific categories, for instance different soil types, land use and elevation zones.
Different catchments were selected globally to best represent each process calibrated (Fig. 3).
Processes were assumed to be linked to different physiographic characteristics (Kuentz et al., 2017)
and catchments with gauging stations where these characteristics were most prominent in the
upstream area were selected (i.e. the representative gauged basin method). For HRUs, separate
calibration was done for the snow-dominated areas (>10% of precipitation falling as snow), as the
snow processes give such strong character to the runoff response and simultaneous calibration with
catchments lacking snow may thus underestimate other flow-controlling processes. The HRUs based
on the ESA CCI 1.6 data was aggregated from 36 classes into 10 (Table 4) for more efficient
calibration and to ensure that some gauged catchments were representing the appointed land-cover.
Some local hydrological features such as large lakes and floodplains were calibrated individually.
When evaluating the effect of this, we discovered some major bias for the Great Lakes in North
America and Malawi and Victoria lakes in Africa. Finally, we introduced the 11[th] step to calibrate the
evaporation of these separately (Fig 3).

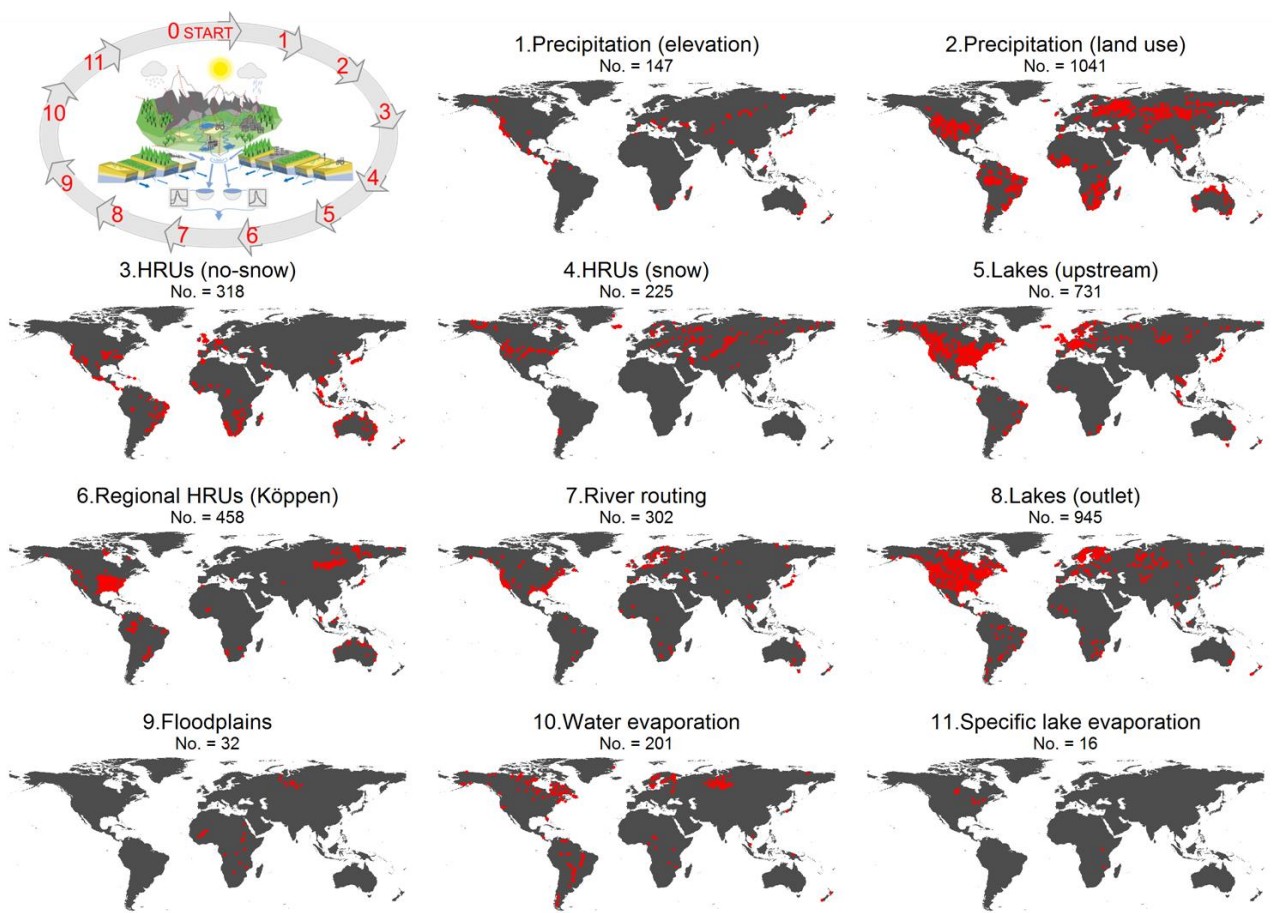

**Figure 3.** Number of gauging stations and their location that was used in each step of the stepwise parameter estimation procedure and evaluation against in-situ observations world-wide.

In total, 6519 river gauges were used for evaluating model performance. Among these, 3656 were used in the calibration but each gauge only affected a few model parameters in the stepwise procedure. Automatic calibration was applied for each subset of parameters and representative catchments in each step, using the Differential Evolution Markov Chain (DEMC) approach (Ter Braak, 2016) to obtain the optimum parameter value in each case. The advantage of DEMC versus plain DE is both the possibility to get a probability-based uncertainty estimate of the global optimum and a better convergence towards it. The DEMC requires several parameters to be fixed and the choice of these parameters was based on a compromise between convergence speed and the accuracy of the resulting parameter set. Global PET parameter values were fixed first, before starting the stepwise procedure, using the MODIS global evapotranspiration product (MOD16) by Mu et al., (2011) for parameter constraints. The parameter ranges were defined as the median and the 3[rd] quartile of the 10% best agreements between HYPE and MODIS in terms of RE. The first selection was done with 400 runs and then repeated for a second round. In addition, a priori parameters (Table A1 in the Appendix) were set for glaciers and soils without calibration, taken from previous applications (e.g. Donnelly et al., 2016; MacDonald et al., 2018). The bare deserts soil was manually calibrated only using 4 stations in the Sahara desert. The area and volume of glaciers were evaluated in 296 glaciers and soil parameters in some 30 catchments. The root zone storage of soils was further calibrated in the parameter setting of each HRU (in step No 4 and 5).

While the calibration period was 1981-2012, it was always preceded by 15 years of initialization.
Different metrics were chosen as calibration criteria, depending on the character of the parameter
and how it influences the model. For instance, Relative Error (RE) was used as a metric in the
calibration of precipitation and PET parameters, since the aim was to correctly represent water
volumes. On the contrary, Correlation Coefficient (CC) was used when the timing was the main goal
(i.e. for river routing or dampening in lakes). If both water volume and timing were required, Kling-
Gupta Efficiency (KGE; Gupta et al., 2009) was used (i.e. for soil discharge from HRUs). Wherever
possible, calibration was made using a daily time-step, while overall model evaluation on the global
scale was made on a monthly time-step.

**Table 4.** Aggregated land covers used for calibrating HRUs, their representation in the upstream catchment and
the number of gauges available for each land cover when estimating parameter values of WWH v1.3.

| Aggregated Land Cover | Original land cover from ESA CCI 1.6 (model HRUs) | Land cover | No. gauges (snow area) | No. gauges (no snow) |
|---|---|---|---|---|
| Bare | Bare areas<br>Consolidated bare areas<br>Unconsolidated bare areas | 35% | 7 | 32 |
| Crop | Cropland, rain fed<br>Herbaceous cover<br>Tree or shrub cover<br>Cropland, irrigated or post-flooding irrigated Rice | 50% | 52 | 30 |
| Grass | Grass | 50% | - | 1 |
| Mosaic | Mosaic cropland (>50%) / natural vegetation (tree, shrub, herbaceous cover) (<50%)<br>Mosaic natural vegetation (tree, shrub, herbaceous cover) (>50%) / cropland (<50%)<br>Mosaic tree and shrub (>50%) / herbaceous cover (<50%)<br>Mosaic herbaceous cover (>50%) / tree and shrub (<50%) | 50% | 39 | 29 |
| Shrub | Shrubland<br>Shrubland evergreen<br>Shrubland deciduous<br>Shrub or herbaceous cover, flooded, fresh/saline/brackish water | 50% | 54 | 17 |
| Sparse | Lichens and mosses<br>Sparse vegetation (tree, shrub, herbaceous cover) (<15%)<br>Sparse shrub (<15%)<br>Sparse herbaceous cover (<15%) | 35% | 40 | 11 |
| TreeBrDecMix | Tree cover, broadleaved, deciduous, closed to open (>15%)<br>Tree cover, broadleaved, deciduous, closed (>40%)<br>Tree cover, broadleaved, deciduous, open (15-40%)<br>Tree cover, mixed leaf type (broadleaved and needle-leaved) | 50% | 26 | 28 |
| TreeBrEvFlood | Tree cover, broadleaved, evergreen, closed to open (>15%)<br>Tree cover, flooded, fresh or brackish water<br>Tree cover, flooded, saline water | 50% | 37 | 30 |

| | | | | |
|---|---|---|---|---|
| TreeNeDec | Tree cover, needle-leaved, deciduous, closed to open (>15%) | 50% | 46 | - |
| | Tree cover, needle-leaved, deciduous, closed (>40%) | | | |
| | Tree cover, needle-leaved, deciduous, open (15-40%) | | | |
| TreeNeEv | Tree cover, needle-leaved, evergreen, closed to open (>15%) | 50% | - | 10 |
| | Tree cover, needle-leaved, evergreen, closed (>40%) | | | |
| | Tree cover, needle-leaved, evergreen, open (15-40%) | | | |
| Urban | Urban | 50% | 21 | 30 |


## 4.3 Model evaluation

The model was evaluated against independent observed river flow by using remaining gauges, which
were not chosen for the calibration procedure. The agreement between modelled and observed
time-series was evaluated using the statistical metric KGE and its components r, $\beta$ and $\alpha$, which are
directly linked with CC (Pearson Correlation Coefficient), RE (Relative Error) and RESD (Relative Error
of Standard Deviation), respectively (Gupta et al., 2009). KGE is defined as:
$$KGE = 1 - \sqrt{(r-1)^2 + (\alpha-1)^2 + (\beta-1)^2} \qquad \text{(Eq. 1)}$$

where:

$$r = CC = \frac{cov(x_o, x_s)}{\sigma_s \sigma_o} \qquad \text{(Eq. 2)}$$

$$\beta = \frac{\mu_s}{\mu_o} \; ; RE = (\beta - 1) \cdot 100 \qquad \text{(Eq. 3)}$$

$$\alpha = \frac{\sigma_s}{\sigma_o} \; ; RESD = (\alpha - 1) \cdot 100 \qquad \text{(Eq. 4)}$$


$x$ represents the discharge time series, $\mu$ the mean value of the discharge time series, and $\sigma$ the
standard deviation of the discharge time series. The sub-indexes $o$ and $s$ represent observed and
simulated discharge time series, respectively. Thus CC represents how well the model dynamics
agree between observations and simulations, i.e. the timing of events but not the magnitude; RE
represents the agreement in volume over time; RESD represents how well the model captures the
amplitude of the hydrograph. KGE was chosen as performance metric to analysis all these aspects
and because it has been found good in capturing both mean and extremes during calibration
(Mizukami et al., 2019). We used the original version so that our results can easily be compared to
other studies reported in the literature, even though non-standard variants may be more efficient
(e.g. Mathevet et al., 2006; Mizukami et al., 2019).
In addition, a number of flow signatures (Table 5) was calculated to explore which part of the
hydrograph is well captured by the model. Flow signatures are used by the catchment modelling
community to condense the hydrological information from time-series (Sivapalan, 2005) and the
choice of flow signatures was guided by previous studies by Olden and Poff (2003) and Kuentz et al.
(2017). In this study, flow signatures were calculated at 5338 gauging stations globally, based on
catchment size and at least 10 years of continuous time-series (see section 2.3).

The model capability in capturing observed flow signatures was then related to upstream physiographical and climatological factors, such as area, mean elevation, drainage density, land-cover, climatic region or aridity index. Catchment modellers tend to study differences and similarities in flow signatures as well as in catchment characteristics to improve understanding of hydrological processes (e.g. Sawicz et al., 2014; Berghuijs et al., 2014; Pechlivanidis and Arheimer, 2015; Rice et al., 2015). In large-sample hydrology it is not possible to examine each hydrograph individually using inspection. As the flow signatures aggregate information about the hydrograph, the model capability to simulate signatures will tell the modeller which part of the hydrograph is better or worse. Linking catchment descriptors to the performance in flow signatures help the modeller to examine whether the process description and model structure are valid across the landscape or if the regionalization of parameter values must be reconsidered for some parts of a large domain. In addition, this exercise will guide the users to judge under which conditions the model is reliable and thus of any use for decision making. In the present study, the physiographic characteristics of catchments were all extracted from the input data files of the WWH version 1.3. For each gauging station with calculated flow signatures, the catchment characteristics were accumulated for all upstream catchments to account for any potential physiographical influence on the flow signal at the observation site (Table 3). Gauging stations were grouped according to the distribution of each physiographic characteristic and model performances in flow signature representation were computed for each of these groups.

**Table 5.** Flow signatures (FS) from observed time-series and physiographic descriptors (T: topography; LC: Land cover; C: climate) from databases in Section 2.1.

| Variable name | Description | Range |
|---|---|---|
| skew (FS) | Skewness = mean/median of daily flows | [0.63 - 70000] |
| MeanQ (FS) | Mean specific flow in mm | [0 - 1024.41] |
| CVQ (FS) | Coef. of variation = standard deviation/mean of daily flows | [0.01 - 46.4] |
| BFI (FS) | Base Flow Index: 7-day minimum flow divided by mean annual daily flow averaged across years | [0 - 0.84] |
| Q5 (FS) | $5^{th}$ percentile of daily specific flow in mm | [0 - 218.04] |
| HFD (FS) | High Flow Discharge: $10^{th}$ percentile of daily flow divided by median daily flow | [0 - 1] |
| Q95 (FS) | $95^{th}$ percentile of daily specific flow in mm | [0 - 2654.81] |
| LowFr (FS) | Total number of low flow spells (threshold equal to 5 % of mean daily flow) divided by the record length | [0 - 1] |
| HighFrVar (FS) | Coef. of Variation in annual number of high flow occurrences (threshold $75^{th}$ percentile) | [0 - 5.48] |
| LowDurVar (FS) | Coef. of Variation in the annual mean duration of low flows (threshold $25^{th}$ percentile) | [0 - 3.78] |
| Mean30dMax (FS) | Mean annual 30-days maximum divided by median flow | [0 - 29.49] |
| Const (FS) | Constancy of daily flow (see Colwell, 1974) | [0.01 - 1] |
| RevVar (FS) | Coef. of variation in annual number of reversals (change in sign in the day-to-day change time series) | [0 - 5.48] |
| RBFlash (FS) | Richard-Baker flashiness: sum if absolute values of day-to-day changes in mean daily flow divided by the sum of all daily flows | [0 - 2] |
| RunoffCo (FS) | Runoff ratio: mean annual flow (in mm $yr^{-1}$) divided by mean annual precipitation | [0 - 1362.52] |
| ActET (FS) | Actual evapotranspiration: mean annual precipitation minus mean annual flow (in mm $yr^{-1}$) | [-100 - 2660.03] |
| Area (T) | Total upstream area of catchment outlet in $km^2$ | [13.5 - 4671536.7] |
| meanElev (T) | Mean elevation of the catchment in m | [3.63 - 5046.16] |
| stdElev (T) | Standard deviation of the elevation of the catchment in m | [1.66 - 1595.89] |
| Meanslope (T) | Mean slope of the catchment | [0 - 224.24] |
| Drainage density (T) | Total length of all streams in the catchment divided by the area of the catchment | [2.19 - 259798.14] |

| 13 land cover variables (LC) | % of the catchment area covered by the following land cover types (see Table XX): Water, Urban, Snow & Ice, Bare, Crop, Mosaic, TreeBrEvFlood, TreeBrdecMix, TreeNeEv, TreeNeDec, Shurb, Grass and Sparse | [0 - 1] |
|---|---|---|
| Pmean (C) | Mean annual precipitation in mm $yr^{-1}$ | [51.5 - 5894.86] |
| SI.Precip (C) | Seasonality index for precipitation: $$SI = \frac{1}{\bar{R}} \cdot \sum_{n=1}^{12} \left| \bar{x}_n - \frac{\bar{R}}{12} \right|$$ $\bar{x}_n$: mean rainfall of month n; $\bar{R}$: mean annual rainfall | [-16.93 - 31] |
| Tmean (C) | Mean annual temperature in degrees | [0.08 - 50.06] |
| AI (C) | Aridity Index: PET/P, where PET is the mean annual potential evapotranspiration and P the mean annual precipitation | [0.05 - 1.28] |
| 5 Köppen regions (C) | % of the catchment area within the following Köppen regions: A (Tropical), B (Arid), C (Temperate), D (Cold-continental) and E (Polar) | [0 - 1] |


# 5. Results


## 5.1 Global river flow and general model performance

To some extent WWH version 1.3 describes hydrological features globally and spatial variability in
factors controlling the runoff mechanisms, although there is still substantial room for improvements
over the coming decade(s). The catchment modelling approach with careful consideration to
hydrography resulted in a new database with delineated hydrographical features (e.g. Fig. 4) of major
importance for hydrological modelling. The merging of several data sources resulted in consistency
between available information on water bodies, topographic data and the river network (e.g. for
glaciers, floodplains, lakes, and gauging stations) so that this information can be used in catchment
modelling and provide results of river flow at a resolution of some 1000 km$^2$ globally.

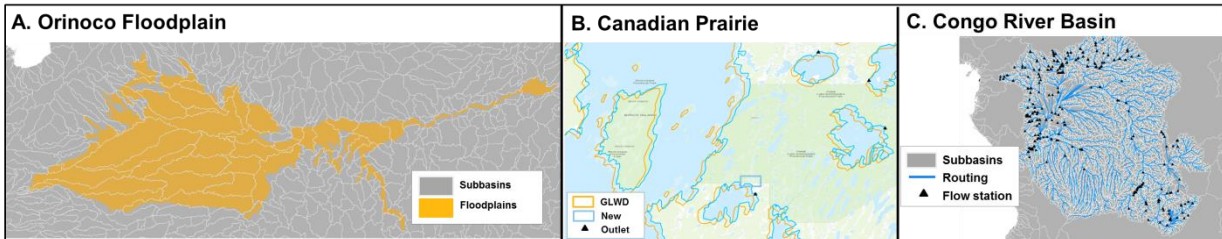

**Figure 4.** Some examples of WWH version 1.3 details in describing hydrography at local and regional scale from
supporting GIS layers: A) subbasins of the Orinocco river defined as a connected floodplain; B) adjustment of
lake areas (New) from merging several data sources (see Section 2.1 and 3.1) and the original GLWD in the
Canadian Prairie; C) river routing and access to flow gauges in the Congo river basin.

The WWH version 1.3 resulted in a realistic spatial pattern of river flow world-wide, clearly
identifying desert areas and the largest rivers (Fig. 5). Compared to other global estimates of average
water flow in major rivers, HYPE gives results in the same order of magnitude, but of course,
comparisons should be based on the same time period to account for natural variability due to
climate oscillations. The Amazon, Congo and Orinocco rivers came out as the three largest ones,

where the river flow of the Amazon river is almost 6 times larger than any other river. Compared to recent estimates by Milliman and Farnsworth (2011), HYPE estimated a higher annual average of river flow in Mississippi, St Lawrence, Amur, and Ob, but less in the rest of the top-ten largest rivers of the world, especially relatively lower values were noted for Ganges-Bahamaputra. For World-Wide HYPE, Yangtze river came out as No 11 and Mekong as No 12, and it should be noted that the river flow to Río de la Plata was separated into Paraná River and Uruguay river (the former ranked as No 13 of the largest rivers).

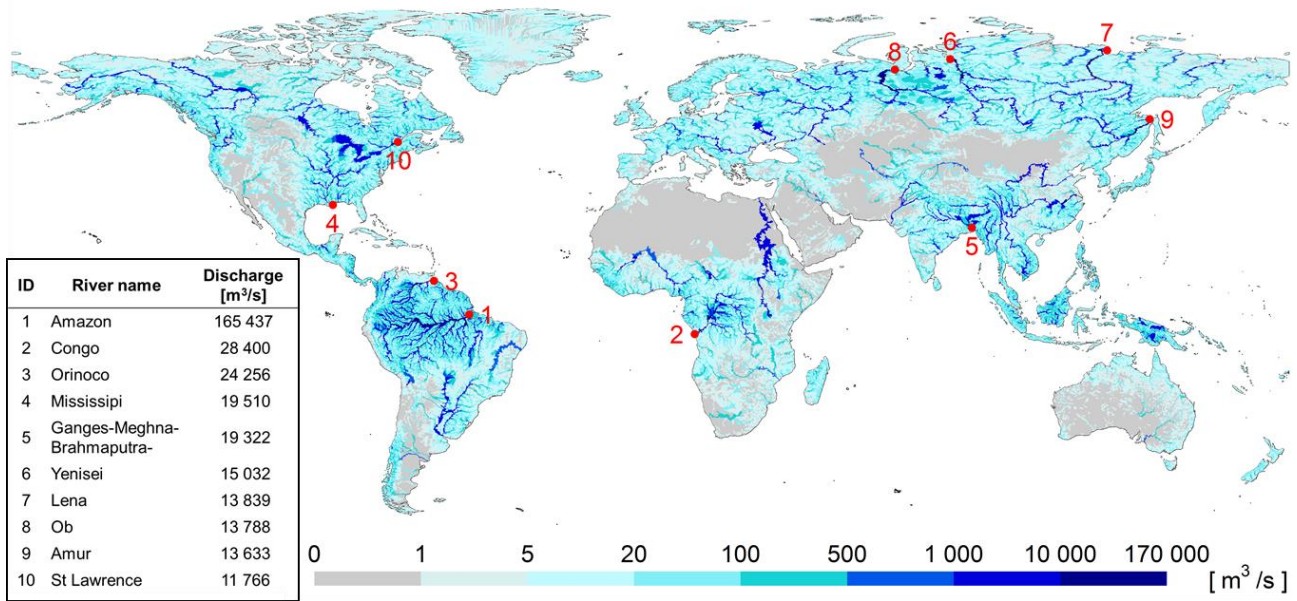

**Figure 5.** Annual mean of river discharge across the globe for the period 1981-2015 estimated with the catchment model WWH version 1.3 (on average 1020 km$^2$ resolution).

On average, for the whole globe and 5338 gauging stations with validated catchment areas and at least ten years of data, the model performance was estimated to a median monthly KGE of 0.40 (Fig. 6). When decomposing the KGE, we found a median correlation coefficient of 0.76 and a median relative error of -15%. This means that the model captures the temporal dynamics of the hydrographs reasonable well in many sites while it generally underestimates the river flow. This underestimation could be resulting from using MODIS when setting calibration ranges. The bluer in Figure 6, the better is the model performance; hence, the model performs best in central Europe, North-East America, Upper Amazon, North Russia (KGE > 0.6). These regions are mostly lowlands and one explanation to good model performance could be that the precipitation from the global meteorological dataset is more correct at lower altitudes with smooth orography. It could also be that the seasonality is more regular and easier to capture.

Model performance was surprisingly similar for the gauges used in parameter estimation and independent ones, with median KGE of 0.41 (2475 stations) and 0.39 (2863 stations), respectively. Among the validation stations, 498 were completely independent without any influence from calibration in any branch of the upstream river network. Also here the model showed similar performance (median KGE = 0.45; median CC = 0.79; median RE = -17). This indicates that the model results are robust and similar model performance can be assumed also in ungauged basins.

If KGE is below -0.41 the model does not contribute with more information than the long-term
average of observations (Knoben et al., 2019), however to judge whether the model performance is
good or bad, the model purpose and use of results must be considered. Most catchment modellers
who come from engineering would probably judge the KGE of 0.40 as poor, but given that global
open input data was used for model setup and rough assumptions were made when generalizing
hydrological processes across the globe, the overall model performance meets the expectations of a
first version.
Global hydrological modellers rarely compare their results to gauged river flow (e.g. Zhao et al.,
2017) but similar results were recently reported when Beck et al. (2016) was testing a scheme for
global parameter regionalization world-wide; in an ensemble of ten global water allocation or land
surface models, the median performance of monthly KGE was found to be 0.22 using 1113 river
gauges for mesoscale catchments globally (median size 500 km$^2$). The best median monthly KGE was
then 0.32 for catchment scale calibration of regionalized parameters, using a gridded HBV model
with a daily time-step globally (Beck, 2016). It is difficult to compare results when not using the same
validation sites or time-period and more concerted actions for model inter-comparison are needed at
this scale. Nevertheless, the catchment modelling approach of the present study seems to have
better performance than other gridded global modelling concepts of river flow (see results from
more models in Beck et al., 2016).
The red spots in Figure 6 indicate where the HYPE model fails (KGE < -1), such as in the US mid-west
(especially Kansas), north-east of Brazil and parts of Africa, Australia and central Asia. When
decomposing the KGE, it was found that the correlation was in general fine. However, the relative
error in standard deviation was causing the main problems showing that the HYPE model does not
capture the variations of the hydrograph, and instead, generates a too even flow. The relative error
also seemed problematic, which indicates problems with the water balance. The model has severe
problems with dry regions and areas with large impact from human alteration and water
management, where the model underestimates the river flow. Such regions are known to be more
difficult for hydrological modelling in general (Bloeschl et al., 2013), but in addition, precipitation
data do not seem to fully capture the influence of topography and mountain ranges. The patterns in
model performance were further investigated in the analysis of model performance versus flow
signatures and physiographic factors (Section 4.3).

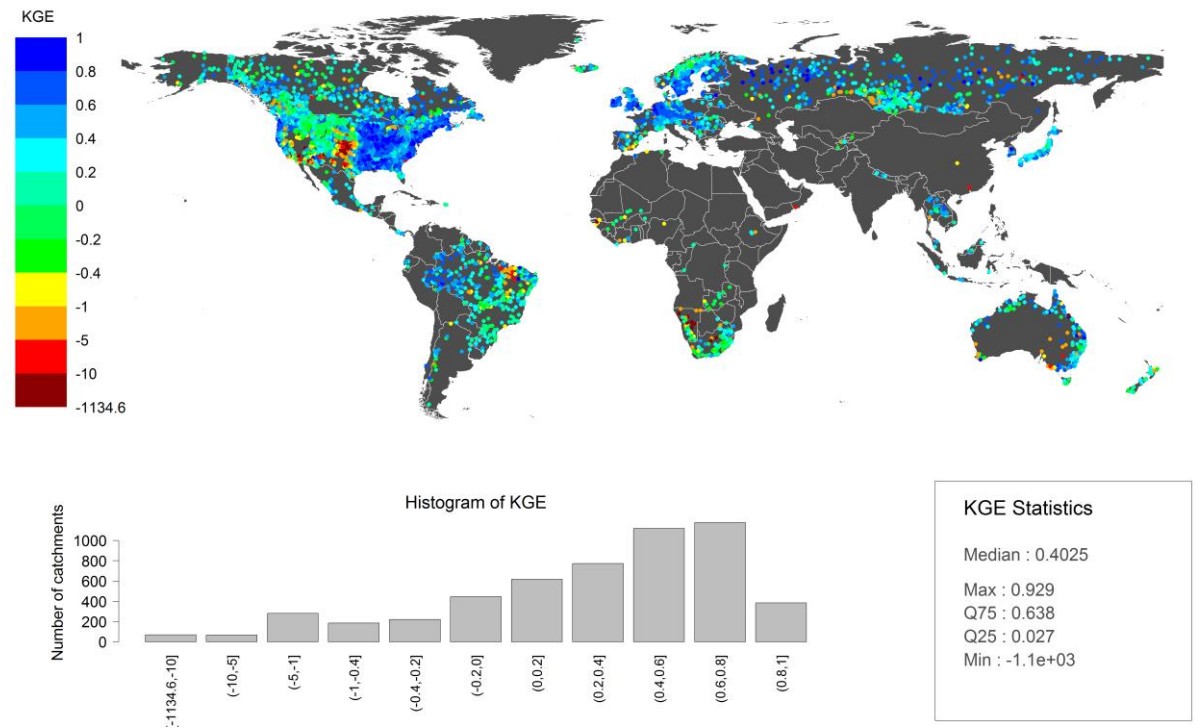

**Figure 6.** Model performance of WWH version 1.3 using the KGE metric of monthly values of ≥ 10 years in each of the 5338 gauging sites for the period 1981-2012. Blue and green indicates that the model provides more information than the long-term observed mean value.

## 5.2 Global parameter values from stepwise calibration

Both model performance in representative catchments and improvement achieved through calibration varied a lot for each hydrological process considered in the stepwise parameter estimation (Table 6). Although, a large number of river gauges was collected for parameter estimation, only a few could be considered as representative with enough quality assurance. More gauges in the calibration procedure would probably have given another result. Nevertheless, the results show promising potential in applying the process descriptions of catchment models also at the global scale.

In spite of the wide spread in geographical locations across the globe, a priori values were reasonable for hydrological processes describing glaciers and soils. As shown in Table 6, the water balance (RE) was improved considerably by first calibrating PET globally, and then precipitation vs altitude of catchment and land-cover type. Simultaneous calibration of soil storage and discharge in HRUs increased the KGE both in areas with and without snow by 0.1 on average. For calibration of river routing and rating curves of lake outflows, the correlation coefficient was used to avoid erroneous compensation of the water balance, as the parameters involved should only set the dynamics of flow and not volume. Especially lake processes benefited from calibration. Less convincing was the metrics from calibration of the floodplains, which were not always improved by the floodplain routine applied. Overall, the results indicate that global parameters are to some extent possible for describing hydrological processes world-wide, using a catchment model and globally available data of

physiographic characteristics to describe spatial variability. Nevertheless, the WWH v.1.3 model has still considerable potential for improvements and to really make use of more advanced calibration techniques, the water balance needs to be improved first as too much volume error makes the tuning of dynamics difficult.

**Table 6.** Metrics of model performance before and after calibrating various hydrological processes simultaneously at a number of selected river gauges, using the stepwise parameter-estimation procedure globally. Parameter values and names in the HYPE model are given in the Appendix.

| Hydrological Process | No. gauges | Median value of metric(s) | |
|---|---|---|---|
| | | Before | After |
| Potential Evapo-Transpiration (3 PET-algorithms: median of ranges constrained with MODIS) | 0 | RE: 11.5 % | RE: 0.5% |
| Glaciers (only evaluated vs mass balance data) | 296 | RE:  0.38% CC:  0.51 | - |
| Soils (average, rock, urban, water, rice) | 25 | RE: -14.1% KGE: 0.2 | |
| Bare soils in deserts (calibrated manually) | 4 | RE: 236.1% | RE: -18.9 |
| 1. Precipitation: catchment elevation | 147 | RE: -6.7% | RE: 4.4% |
| 2. Precipitation: land-cover altitude | 1041 | RE: 24.3% | RE: 10.1% |
| 3. HRUs in areas without snow | 318 | KGE: 0.16 | KGE: 0.27 |
| 4. HRUs in areas with snow: ET, recession and active soil depth | 225 | KGE: 0.16 | KGE: 0.24 |
| 5. Upstream lakes | 731 | CC: 0.71 | CC: 0.72 |
| 6. Regionalised ET (in 12 Köppen climate regions) | 458 | KGE: 0.58 | KGE: 0.62 |
| 7. River routing | 302 | CC: 0.70 | CC: 0.71 |
| 8. Lake rating curve | 945 | CC: 0.50 | CC: 0.59 |
| 9. Floodplains (partly calibrated manually) | 32 | KGE: -0.03 | KGE: 0.03 |
| 10. Evaporation from water surface | 201 | RE: -20.7% | RE: -12.2% |
| 11. Specific lake evaporation | 16 | RE: 24.8% | RE: 4.8% |

## 5.3 Model evaluation against flow signatures

The WWH1.3 is more prone to success or failure in simulating specific flow signatures than to specific physiographic conditions, which is visualized by vertical rather than horizontal stripes in Figure 7. In general, the model shows reasonable KGE and CC for spatial variability of flow signatures across the globe (i.e. a lot of blue in the two panels to the left in Fig. 7). However, the RE and the standard deviation of the RE (RESD) are less convincing (i.e. the two panels to the right). This means that the model can capture the relative difference in flow signature and the spatial pattern globally, but not always the magnitudes, nor the spread between highest and lowest values. The relative errors are mostly due to underestimations, except for skewness, low flows and actual potential evapotranspiration; the two latter are always over-estimated when not within ±25% bias. Overall,

the model shows good potential to capture spatial variability of high flows (Q95), duration of low
flows (LowDurVar), monthly high flows (Mean30dMax) and constancy of daily flows (Const). These
results were found robust and independent of metrics or physiography. The results implies that the
overall process understanding behind the HYPE model structure and the assumptions of catchment
similarities in the set-up may be relevant at the global scale, but that the estimation of parameter
values or quality of forcing data are not optimal for capturing the flow dynamics.
The model shows most difficulties in capturing skewness in observed time-series (skew), the number
of high flow occurrences (HighFrVar), and base flow as average (BFI), or absolute low flows (Q5).
Short-term fluctuations (RevVar and RBFlash) are also rather difficult for the model to capture. Some
results are not consistent between metrics; for coefficient of variation (CVQ) the RE was good while
the RESD was poor. This indicates that the model does not capture the amplitude in variation
between sites even if the bias is small. The opposite was found for high flow discharge (HFD) and
low-flow spells (LowFr), i.e. poor performance in volumes but RESD showing that the variability is
captured.
For the remaining flow signatures studied, it was interesting to note that the model performance
could be linked to physiographic characteristics, indicating that the model structure and global
parameters are valid for some environments but not for others. For instance, the volume of mean
specific flow (RE of MeanQ) is especially difficult to capture in regions with needle-leaved, deciduous
trees (TreeNeDec) and for medium and large flows in the Köppen region B (Arid), large flows in D
(Cold-continental) and small flows in E (Polar). Moreover, the analysis shows that the model tends to
fail with the mean flow in catchments with high elevation, high slope, small fraction water and urban
land-cover, and little or much of snow and ice. This shows where efforts need to be taken to improve
the model in its next version.
For other water-balance indices, it was interesting to note that the ratio between precipitation and
river flow (RunoffCo) show good results (RE ± 25%) all over Köppen region C (Temperate) but
otherwise is often underestimated for some parts of the quartile range of physiographic variables
studied. On the contrary, precipitation minus flow (ActET) is over-estimated in parts of the quartile
range, except for the good results in Köppen region C, needle-leaved, deciduous trees (TreeNeDec)
and regions with snow and ice (i.e. where mean specific runoff failed). Figure 7 clearly shows the
compensating errors between processes governing the runoff coefficient and actual
evapotranspiration, with one being over-estimated when the other is underestimated for the same
specific physiographic conditions. This indicates the need for recalibrating the HRUs of WWH in its
next version, but also reconsidering the initial parameters for evapotranspiration and the quality of
the precipitation grid and its linkage with the catchments. It is rather common to use Köppen when
evaluating ET (e.g. Liu et al, 2016) but it may not be the best separator hydrologically (Knoben et al.,
2018) so model performance should preferably be evaluated and calibrated in clusters based on
other characteristics in the future.

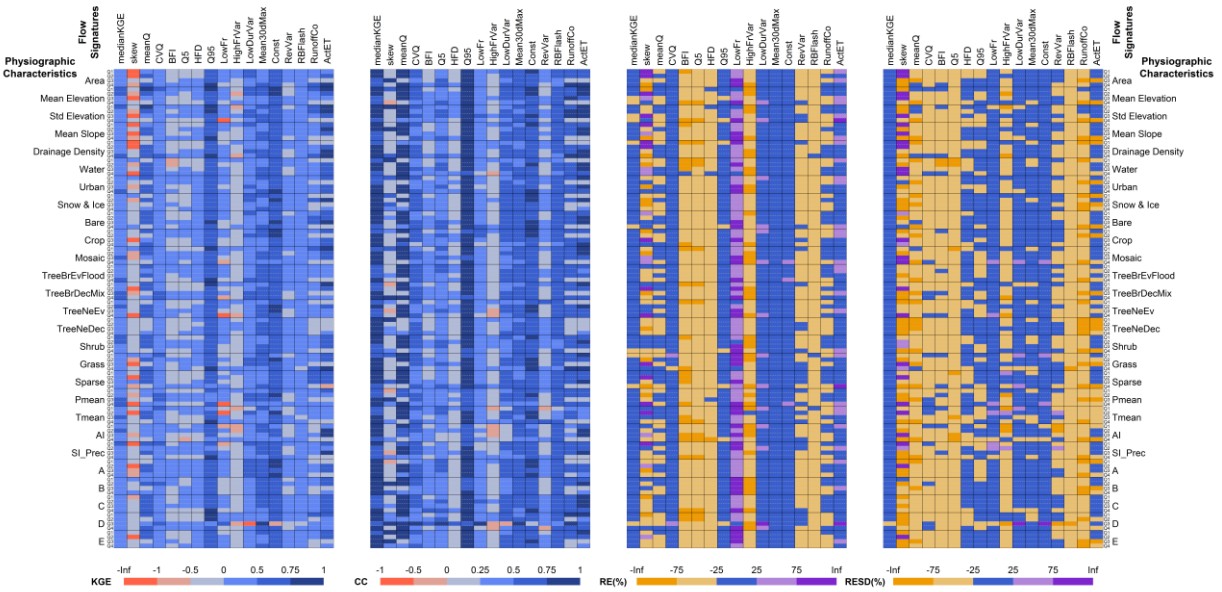

**Figure 7.** Matrix showing the relation between model capacity to capture flow signatures (colors, where blue is good and yellow/red/purple is poor performance) and physiography of catchments, divided into quartiles (Q1-Q4) for characteristics of the total area upstream each gauging station with more than 10 years of continuous data (5338 catchments). Description of flow signatures and physiographic characteristics are found in Table 4-5 and metrics used for model performance in Eq. 1-4.

## 6. Discussion

This experiment of whether it is now possible and timely to apply catchment modelling techniques to advance global hydrological modelling gave some diverse results. Regarding physiographic data, it is now possible to delineate catchments thanks to high-resolution topographic data (Yamazaki et al., 2017) and there are many global datasets readily available with necessary physiographic input data for catchment modelling also including local hydrological features and waterbodies (e.g. sinks and floodplains) that are normally not included in the traditional global models (e.g. Zhao et al., 2017). Nevertheless, before merging the databases we found that they need to be harmonized and quality assured, which has already been noted in previous studies (e.g. Kauffeldt et al., 2013). For meteorological data, global precipitation from re-analysis products are well known to contribute a lot to the output uncertainty in traditional global modelling (e.g. Döll and Fiedler, 2008; Biemans et al., 2009) and this was still the case when applying catchment modelling; although the precipitation grid was bias-adjusted against observations (Berg et al., 2018) and further adjusted with elevation during calibration, the density of stations at the global scale was not sufficient for the resolution of catchments. New high-resolution products from the meteorological community have potential to become a game-changer in global hydrological modelling.

The test whether parameter estimation methods from the catchment modelling community could improve model performance in global hydrological predictions resulted in better metrics than previously reported by e.g. Beck et al. (2016). Despite the large sample of river gauges, however, we

experienced that it was not distributed well enough to cover the large domain. Screening of the
gauged data quality showed that most regions worldwide have access to some high-quality time
series of river flow (Crochemore et al., 2019) but for the stepwise procedure applied here this was
still not enough for many of the pre-defined calibration steps. Even when merging the original ESA
land cover classes before calibration (Table 4) sufficient gauged data was missing. As the structure of
the catchment model reflects the modellers' process understanding and as parameters must be
estimated (Wagener, 2003) a better compromise must be made between the HYPE structure or set-
up and flow gauges available for the global calibration scheme. Hence, the ecosystem approach
needs to be elaborated with better defined clusters for catchment similarity across the globe to be
truly helpful at this scale.
With current computational resources it was possible to use automatic iterative calibration
techniques from the catchment community (i.e. DEMC, Ter Braak, 2016) to obtain the optimum
parameter values from several iterations, also across large samples of gauges. However, enough
computational resources were still lacking for advanced uncertainty analysis, such as using the GLUE
(Beven and Binley, 1992).
To sum up, we found that the catchment model application at global scale could be considered
timely because it was doable and now there is potential for improvements, although, even at this
stage the model might be useful for some purposes in some regions, as discussed below.

**6.1 Potential for improvements**
The results from evaluating model performance using several metrics, several thousand gauges and
numerous flow signatures, gave clear indication on regions where the model most urgently needs
improvements. A thorough analysis would also benefit from evaluation against independent data of
spatial patterns of hydrological variables, for instance from Earth Observations. In general, the WWH
model has severe problems with dry regions and base flow conditions where the flow is sporadic
(e.g. red areas in Fig. 5). The flow generating processes in such areas are known to be difficult to
model (Bloeschl et al., 2013). For instance, most model concepts, and also the WWH, have problems
with the great plains of US (e.g. Mizukami et al., 2017; Newman et al., 2017), where the terrain is
complex with prairie potholes, which are disconnected from the rivers, and precipitation comprise a
major source of hydrologic model error (e.g. Clark and Slater, 2006). Poor model performance were
also found for the tundra and deserts, but it should then be recognized that the parameters for these
regions were estimated using only four time-series for bare soils (Table 6); including more gauging
stations would be a way to improve the model here. In large parts of Africa, however, model errors
could be linked to the soil-runoff parameters and local calibration based on catchment similarities
has already been found to improve the performance a lot in West Africa.
In the snow-dominated part of the globe, extensive hydropower regulation change the natural
variability of river discharge (Déry et al., 2016; Arheimer et al., 2017) but the global databases miss
out of all medium and small dams that may affect discharge along these river networks. A general
problem with modelling river regulation is that reservoirs can have multi-purposes and must be
examined individually to understand the regulation schemes applied. Such analyses have started and
shown potential to improve the global model a lot as the poorest model results are often linked to
river regulations. However, individual reservoir calibration will be very time-consuming, so instead,
we suggest starting with improvements that can be undertaken relatively quickly and easily. These
mainly focus on the overall water balance. Firstly, the global water balance can be improved through
re-calibration but some basic concepts need to be adjusted accordingly: (i) more careful analyses
indicate that the choice of climate regions based on Köppen's classification for applying the different
PET algorithms was not optimal and needs some adjustments, (ii) linking the centroid of the
catchments to the nearest precipitation grid seems to remove a lot of the spatial variation and
instead an average of nearest grids should be tried. Secondly, the HRUs can be recalibrated and
reconsidered, and we suggest (i) testing a calibration scheme based on regionalized parameters
rather than global, using clustering based on physiographic similarities (e.g. Hundecha et al., 2016),
(ii) including soil properties in the HRU concept again (as in the original version of HYPE, see
Lindström et al., 2010) to account for spatial variability in soil-water discharge linked to porosity in
addition to vegetation and elevation. Thirdly, the behaviour of hydrological features, such as lakes,
reservoirs, glaciers, and floodplains can be evaluated and calibrated separately, after categorizing
them more carefully or from individual tuning. Finally, more observations can be included, both in-
situ by adding more gauges to the system and from global Earth Observation products, for instance
on water levels and storage. Hence, each step in Fig. 3 still has potential for model improvements.
The stepwise parameter-estimation approach should ideally be cycled a couple of times to find
robust values under new fixed parameter conditions. However, as the model was carefully evaluated
during the calibration, there were a lot of bug fixing, corrections and additional improvements
resulting between the steps and time was rather spent on this than on several full-filled iterations.
Therefore, the stepwise calibration was subjected to several re-takes and shifts between steps until it
eventually could full-fill all the calibration steps in one entire sequence (Fig. 8). Hence, only one loop
was done for parameter estimations in this study. The procedure was judged as very useful for the
model to be potentially right for the right reason, but also very time-consuming. However, applying a
catchment modeller's approach, this is inevitable for reliably integrated catchment modelling and
both the stepwise calibration and iterative model corrections will continue with new model versions.

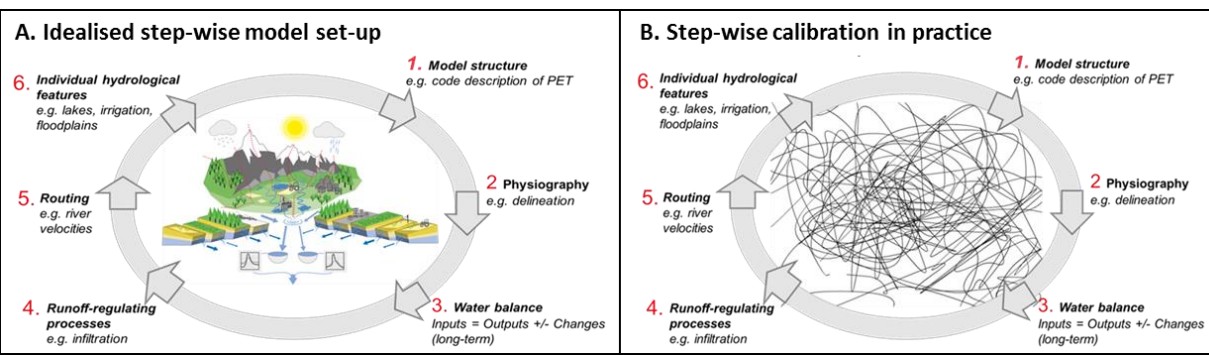

**Figure 8.** Discrepancy between the idealised procedure for stepwise calibration (A) and the numerous
iterations between the steps that appear in reality (B), leading to overall model corrections.

Another important next step in model evaluation and improvement would be to initiate a concerted
model inter-comparison study at the global scale with benchmarking (e.g. Newman et al., 2017), as
we currently lack such studies for global modelling of river flow. Focus should then be on comparing
model performance in general but also on input data and performance of specific hydrological
processes to understand differences between various model concepts. The latter could be done by
using the representative gauged basin approach, as in this study, to evaluate model performance for
sites where flow is dominated by certain processes or by analysing specific parts of the hydrograph
(or flow signatures) that represents time periods when specific processes dominate the flow
generation. In addition to river gauges, other data sources should be used for model evaluation of
spatial patterns, e.g. earth observations. Specific areas that are intensively managed and impacted by
humans should also be distinguished and evaluated separately to better understanding process
variability vs human impacts. Various sources of input data (from which errors may propagate)
should also be evaluated to improve global hydrological modelling.

## 6.2 Model usefulness

Catchment models are often applied by water managers and the usefulness is part of the concept,
however, to provide global hydrological data that is relevant locally is far from trivial (e.g. Wood et
al., 2011; Bierkens et al., 2015). The result analysis of this first version of the WWH model shows that
it can only to some extent be useful for water managers in some regions globally. For instance, long-
term averages are rather reliable in Eastern USA, Europe, South-East Asia, Japan as well as most of
Russia, Canada, and South America. Here the model could thus be used for e.g. analysing shifts in
water resources between different climate periods. For high flows, monthly values show good
performance as well as the spatial pattern of relative values. This implies that the model could be
used for seasonal forecasting of recharge to hydropower reservoirs, for which these variables are
often used. Accordingly, the model has already been applied for producing water-related climate
impact indicators and it is set-up operationally to provide monthly river-flow forecasts for 6 months
ahead (http://hypeweb.smhi.se/).
In many areas, HYPE should still be considered as a scientific tool and cannot be used locally by water
managers because of poor performance. However, the model provides a first platform for catchment
modelling to be further refined and experimented with at the global, regional and local scales. Parts
of the model can be extracted (e.g. specific catchments or countries) and used as infrastructure,
when starting the time-consuming process of setting up a catchment model. The model can then be
improved for the selected catchments by exchanging the global input data with local data and
knowledge, as well as parameters estimated to fit with local observations. Significant improvements
in model performance from such a procedure have already been noted for West Africa (Andersson et
al., 2017).
In Sweden the operational HYPE model runs with national data and adjusted parameter values,
providing an average daily NSE (Nash and Sutcliffe, 1970) of 0.83 for 222 stations with ≤5%
regulation and an average relative volume error of ±5% for the period 1999–2008. For all gauging
sites (some 400) with both regulated and unregulated rivers, the mean monthly NSE is 0.80. The

Swedish HYPE model also started with poor performance in its first version, but has been improved incrementally during more than 10 years and has proven very useful in providing decision-support to society. It supports a national warning service with operational forecasting of floods and droughts (e.g. Pechlivanidis et al., 2014), and the water framework directive for measure plans to improve water quality (e.g. Arheimer et al., 2015). Moreover, it has been used in assessments of hydro-morphological impact (e.g. Arheimer and Lindström, 2014), climate-change impact analysis (e.g. Arheimer and Lindström, 2015) and combined effects from multiple-drivers on water resources in a changing environment (e.g. Arheimer et al., 2017; Arheimer et al., 2018; Arheimer and Lindström, 2019).

Thus, it is found very useful to have a national multi-catchment model to support society in water related issues. This should be encouraging for other countries who do not yet have a national model set-up and also for international river basin authorities searching for a more harmonized way to predict river flow across administrative borders. Using the WWH as a starting point would be a quick and low-cost alternative for getting started with more detailed catchment modelling for decision-support in water management. Parts of the model are therefore shared and can be requested at http://hypecode.smhi.se/. Using a common framework for catchment modelling by many research groups and practitioners will probably advance science as it enables a critical mass and better communication when sharing experiences. Only when using the same methods or data, there is full transparency in the research process so that scientific progress and failures can be clearly understood, shared and learnt from. The WWH could be one stepping stone in such a collaborative process between catchment modellers across the globe. Therefore, SMHI annually offers a free training course since 2011, accompanied with travel grants for participants from developing countries since 2013. Every year about 30 new persons are trained in HYPE and get access to a piece of the modelled world, resulting in model refinements and various regional assessments around the globe e.g. climate-change impact on Hudson Bay (MacDonald et al., 2018), flow forecasts in Niger River (Andersson et al., 2017), hydromorphological evolution of Mackenzie delta (Vesakoski et al., 2017), and water quality in South Africa (Namugize et al., 2017) or England (Hankin et al., 2019).

## 7. Conclusions

This study shows the usefulness of applying catchment modelling methods (topographic catchment delineation, stepwise calibration, performance evaluation against a large sample of observations using several metrics and flow signatures) to help advance global hydrological modelling. The catchment modelling approach resulted in better performance (median monthly KGE = 0.4) than what has been reported so far from more traditional gridded modelling of river flow at the global scale. Major variability in hydrological processes could be recognized world-wide using global parameters, as these were linked to physiographical variables to describe spatial variability and calibrated in a stepwise manner. Clearly, the community of catchment modellers' can contribute to research also at the global scale nowadays with the numerous open data available and advanced processing facilities.

However, the WWH resulting from this first model version should be used with caution (especially in
dry regions) as the performance may still be of low quality for local or regional applications in water
management. Geographically, the model performs best in Eastern USA, Europe, South-East Asia and
Japan, as well as parts of Russia, Canada, and South America. The model shows overall good
potential to capture flow signatures of monthly high flows, spatial variability of high flows, duration
of low flows and constancy of daily flow. Nevertheless, there remains large potential for model
improvements and it is suggested both to redo the calibration and reconsider parts of the model
structure for the next WWH version.
The stepwise calibration procedure was judged as very useful for the model to be potentially right for
the right reason, but also very time-consuming and data demanding. The calibration cycle is
suggested to be repeated a couple of times to find robust values under new fixed parameter
conditions, which is a long-term commitment of continuous model refinement. The model set-up will
be released in new model versions during this incremental improvement. For the next version,
special focus will be given to the water balance (i.e. precipitation and evapotranspiration), soil
storage and dynamics from hydrological features, such as lakes, reservoirs, and floodplains.
The model is shared by providing a piece of the world to modellers working at the regional scale to
appreciate local knowledge, establish a critical mass of experts from different parts of the world and
improve the model in a collaborative manner. The model can serve as a fast track to a model
environment for users who do not have this ready at hands and in return the WWH can be improved
from feedback on hydrological processes from local experts across the world. Potentially it will
accelerate scientific advancement if more researchers start using the same tools and data, which
makes it easier to be transparent when evaluating and comparing scientific results. SMHI commits to
long-term management, continuous refinement, supporting tools, training and documentation of the
WWH model.

# Code availability

http://hypecode.smhi.se

# Data availability

http://hypeweb.smhi.se

# Appendix


The Table below show additional information to Table A1 regarding which HYPE parameters that
were calibrated for each process during the model set-up and the range of resulting parameter
values. Description of each parameter can be found in the HYPE wiki at http://hypeweb.smhi.se/.

**Table A1.** Metrics and parameter values from the stepwise parameter-estimation globally. Parameter names and values are given in the same order of appearance (columns 2 and 6).

| Hydrological Process | HYPE parameters http://hypecode.smhi.se/ | No. gauges | Median value of metric(s) Before | After | Parameter value(s) |
|---|---|---|---|---|---|
| Potential Evapo-Transpiration (3 PET-algorithms: median of ranges constrained with MODIS) | Jhtadd, jhtscale, kc2, kc3, kc4, krs, alb, alfapt | 0 | RE: 11.5 % | RE: 0.5% | 5; 100; [0.7-1.7]; [0.15-1.7]; [0.8-1.6]; 0.16; [0.3-0.8]; 1.26 |
| Glaciers (only evaluated vs mass balance data) | glacvexp, glacvcoef, glacvexp1, glacvcoef, glac2arlim, glacannmb, glacttmp, glaccmlt, glaccmrad, glaccmrefr, glacalb, fepotglac | 296 | RE: 0.38% CC: 0.51 | - | 1.38, 0.17 1.25, 12.88 25 000 000, 0, 0, 1.58, 0.19, 0.06, 0.35, 0 |
| Soils (average, rock, urban, water, rice) | 5 soils: rrcs1, rrcs2, rrcs3,trrcs, mperc1, mperc2, macrate, mactrinf, mactrsm, srrate, wcwp1-3, wcfc1-3, wcep1-3 | 25 | RE: -14.1% KGE: 0.2 | | Ranges: [0.20 - 0.5]; [0.01 - 0.45]; [0.01 - 0.1]; [0.05 - 0.35]; [30 – 100]; [10 - 60]; [0.05 – 0.7]; [12 - 30]; [0.3 – 0.9]; [0.01 – 0.3]; [0.01 – 0.6]; [0.2 – 0.6] ; [0.01 – 0.5] |
| Bare soils in deserts (calibrated manually) | rrcs1, rrcs2, rrcs3, trrcs, mperc1 mperc2, macrate, mactrinf, mactrsm, sfrost, srrate, wcwp1-3, wcfc1-3, wcep1-3 | 4 | RE: 236.1% | RE: -18.9 | 0.6, 0.3, 0.0002, 0.15, 10, 0.1, 10, 0.8, 1, 0.01, 0.01, 0.0001, 0.0001, 0.3, 0.3, 0.0001, 0.03, 0.03, 0.0003 |
| 1. Precipitation: catchment elevation | Pcelevth, Pcelevadd, Pcelevmax | 147 | RE: -6.7% | RE: 4.4% | 500; 0.01; 0.7 |
| 2. Precipitation: land-cover altitude | 5 elevation zones: pcluse | 1041 | RE: 24.3% | RE: 10.1% | 0.05; 0.2; 0.25; 0.25; 0.35 |
| 3. HRUs in areas without snow | 10 HRUs: kc2, kc3, kc4, alb, soilcorr, srrcs, soilcorr | 318 | KGE: 0.16 | KGE: 0.27 | Range: [0.90-1.54]; [0.40-1.77]; [0.20-1.90]; [0.20-0.80]; [1.00-10.55]; [0.03-0.50]; |

| | | | | | |
|---|---|---|---|---|---|
| 4. HRUs in areas with snow: ET, recession and active soil depth | 10 HRUs: ttmp, cmlt, cmrad, fscdist0, fepotsnow | 225 | KGE: 0.16 | KGE: 0.24 | Ranges: [-2.67-1.80]; [1.10-4.00]; [0.16-1.5]; [0.20-0.75]; [0.09-0.98] |
| 5. Upstream lakes | Ilratk, ilratp | 731 | CC: 0.71 | CC: 0.72 | 1.8; 1.4 (depth: 5 m; icatch: 0.3) |
| 6. Regionalised ET (in 12 Köppen climate regions) | 12 climates: cevpcorr | 458 | KGE: 0.58 | KGE: 0.62 | Ranges: [-0.43 – 0.38] |
| 7. River routing | rivvel, damp | 302 | CC: 0.70 | CC: 0.71 | 0.6; 1.0 |
| 8. Lake rating curve | 888 Lakes: rate; exp (LakeData.txt) | 945 | CC: 0.50 | CC: 0.59 | Ranges: [0.001– 1013]; [1.002 – 3.0]; |
| 9. Floodplains (partly calibrated manually) | 13 Floodplains: rclfp; rclpl; rcrfp; rcfpr (FloodData.txt) | 32 | KGE: -0.03 | KGE: 0.03 | Ranges: [0.05 – 0.99]; [0.15 – 0.90]; [0.05 – 0.99]; [0.15 – 0.90] |
| 10. Evaporation from water surface | $kc2_{water}$, $kc3_{water}$, $kc4_{water}$ | 201 | RE: -20.7% | RE: -12.2% | 1.36; 0.65; 1.25 |
| 11. Specific lake evaporation | 2 regions: cevpcorr | 16 | RE: 24.8% | RE: 4.8% | Ranges: [0.375-0.5] |


# Acknowledgements

We would like to thank all data providers listed in Table 1-3 who make their results and observations readily available for re-purposing; without you any global hydrological modelling would not be possible at all. Especially we would like to express our gratitude to Dr. Dai Yamazaki, University of Tokyo, for developing and sharing the global width database for large rivers, which we found very useful. The WWH was developed at the SMHI Hydrological Research unit, where much work is done in common taking advantages from previous work and several projects running in parallel in the group. It was indeed a team work. We would especially like to acknowledge contributions from our colleagues Jörgen Rosberg, Lotta Pers, David Gustafsson and Peter Berg, who provided much of the model infrastructure. Time-series and maps from the World-Wide HYPE model are available for free downloading at http://hypeweb.smhi.se/ and documentation and open source code of the HYPE model is available at http://hypecode.smhi.se/.

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
