# Peer review of "Global catchment modelling using World-Wide HYPE"

_Hydrology and Earth System Sciences, 2019_

## Referee Comment (RC1) · Anonymous Referee #1 · 9 May 2019

The discussion paper represents a very impressive body of research that is well-summarized given the amount of information to be documented for the global application of a catchment model. The use of this global model to then gain insights into global water availability is an additional important contribution only achievable through the thoughtful and comprehensive modeling work presented here.

I also want to make note of the well-referenced datasets used in this modeling effort that are nicely documented in the form of several tables and will provide an excellent reference for readers that have an interest in catchment modeling at the global domain. Lastly, I commend the the authors on their candid discussions regarding the model

performance, parameter estimation, data challenges, and limitations, which will serve to identify the critical research paths moving forward to improve catchments modeling at the global domain.

My comments for improvement mainly lie in providing additional detail on the methods and justification for choices in the modeling process. In some places, the language is feels rushed and I hope my comments also address this point.

Line 95-96: Define what is meant by a "multi-catchment approach for a large domain"

Line 199: Consider emphasizing here the difference and advantages of catchment-scale modeling and other types of models so the importance and significance of catchment based models at the global domain is fully understood.

It might be more logical to the flow of the paper to make Section 3.2 into Section 2, meaning to introduce the model first and then discuss the data and methods used to develop and evaluate the model.

Lines 136-143: How did resolve any spatial differences with the areas not covered by the higher resolution dataset

Line 148-150: How did you identify these flood risk areas?

Change the title of Section 2.2 to "Climate Data" and Section to 2.3 to "Hydrologic Data"

Table 3: Are all time series available at a daily time step?

Lines 202-203: Are versions 1.0 and 1.2 published and citable?

Lines 210-211: How many catchments fell into the category of needing a longer initialization? Was it 10% (100% minus the 90% mentioned in line 209)? This is not clear. How was the screening done to pick these catchments needing a longer initialization?

Line 221: Is the WHIST also new to this paper or should there be a reference here?

Line 271: Change "percolating" to "percolated"

Lines 275-276: Change to read: "encompassess a large number of sinks due to climate and topography and there existed a national...."

Lines 284-288: How were these steps 1-3 done? More detail needs to be provided here.

Line 294-298: Can you offer some specific justification for this assignment was done for soil depth?

Line 318 could read "fluxes well relate to vegetation...conditions rather than soil..."

Line 324: Should read "while a gladter routine accounts for"

Line 326: Start new paragraph at the word "There"

Section 3.3 and Figure 3: Could you add more detail about how the process used to group catchments?

Line 367: Why was the DEMC preferred over other approaches?

---

## Referee Comment (RC2) · Anonymous Referee #2 · 25 Jun 2019

Overall opinion:

Overall, this is a useful synthesis to all those who want to know the state of development of the HYPE model. The only problem I see is in the overly emphatic statements : I have no problem with publishing modest modelling results, or even complete model failures, provided modest results are called modest, failures are named failures. A mean KGE of 0.4 reflects a poor fit, but it may be the most that such a worldwide approach can bring. At this scale, we lack references. In the "model usefulness" section especially you should try to be more modest. Start by stating that in many areas HYPE should still be considered as a scientific tool, and that it cannot be useful

to managers because of its poor performances. Only for a small percentage (10% or 20 % ?), it is usable by a manager. Don't forget that managers have always high model efficiency expectations.

Technical suggestions:

1. To be able to judge of the fit quality, we would have needed something like a classical (let's say HBV) lumped hydrological model applied as reference, in calibration mode, or even in ungagged mode with a single parameter set for the entire globe.

2. I am not convinced by your introduction much too long, not really informative, a lot of commonplace statements. You cite all the "politically correct" papers of the moment, but you could go straight to the point: you are a recognized group with a first worldwide application of your model. That's all.

3. Using KGE is OK, but I would suggest to use a bounded version (between -1 and 1, see Mathevet et al. 2006) because we are not interested by large negative values.

References

Mathevet, T., Michel, C., Andréassian, V. & Perrin, C., 2006. A bounded version of the Nash-Sutcliffe criterion for better model assessment on large sets of basins. IAHS Red Books Series n°307, pp. 211-219.
* * *

---

## Author Comment (AC1) · 31 Aug 2019

Dear Editor,

We appreciate that the manuscript was well received but needed some improvements. Please, find our responses below to the suggestions and questions by the referees. A new version of the manuscript is now available.

In addition we had some comments by email from Dr. Hylke Beck, which we have also included in the revision of the manuscript.

**Response to Editors comments:**

*vague terms being used about what is 'successful';

The term 'successful' was used in three places in the text and has now been replaced as follows:

(1) In the abstract to: 'more useful results for water management'
(2) In the Result section 5.1: (removed)
(3) In the Discussion section 6.1 to: 'eventually'

*about the concepts of what is preserved in the model in terms of mass balance losses or gains in terms of fitting signals;

We are fitting the parameters but the mass (water balance) is conserved in the HYPE model concept. We have now specified this better in the model description (Chapter 2). We also included the following sentence with reference to a more detailed description of all algorithms in the code: "Parameter values regulate the fluxes between water storages in the landscape and interaction with boundary condition of the atmosphere and deep ground water aquifers (see detailed model documentation at hypeweb.smhi.se)."

*a discussion of how models can be compared across the GRDC and all the human impacts and the implications that has for understand process variability vs human impacts;

Thanks for this comment; it is truly a very important next step in model evaluation and improvement. We included this aspect as follows:

First, in Chapter 5 (Results) the WWH model results are compared with previous modelling results by Beck et al., 2016, who used the HBV model in selected basins across the globe. The text mentions the difficulties when comparing model results only using the literature as different river gauges and time-periods are being used, and we now also added a text explicitly saying that "…. more concerted actions for model inter-comparison are needed at this scale." (section 5.1)

Second, in Chapter 6 (Discussion), we have now added a paragraph at the end of section 6.1 (potential for improvements) discussing the importance of model inter-comparison and how to design such a study to better understanding process variability vs human impacts. In fact, such an initiative was already taken in a splinter session at EGU 2019 for the continental scale (Europe), but should be extended to also include the global scale.

*understanding and interpreting the KGE values.

We have now extended the text to better explain the interpretation of the KGE components in the model evaluation section 3.4, where the KGE is explained.

*I also feel that there needs to be a better evaluation as to why certain regions behave well or not in terms of increasing a scientific interpretation in the paper to be novel for publication…

First paragraph in discussion section 6.1 is now extended with two paragraphs describing more in detail the problems and potential for improvements in certain regions. We agree that this improve the scientific value of the paper. The spatial analysis is now organised as follows:

The spatial patterns and discrepancy in model performance are described in the Result sections 5.1. (especially the 4[rd] paragraph) and in 5.3.

In the Discussion section 6.1 we discuss major potentials for improvements to capture processes better in regions where the model fails. This section is now extended with examples of interpretations and references to other studies and on-going work in model evaluation.

A more thorough analysis would benefit from evaluation against independent data of spatial patterns of hydrological variables, for instance from Earth Observations. This is now mentioned in the text. In fact, such an extensive analysis has been performed, but is too extensive to be included here and will be described in another paper, soon to be submitted for scientific publication to continue the diagnosis of the model concept. We chose not to refer to this study yet, as we don't know when (or where) it will be published.

**Response to Anonymous Referee #1**

The discussion paper represents a very impressive body of research that is well summarized given the amount of information to be documented for the global application of a catchment model. The use of this global model to then gain insights into global water availability is an additional important contribution only achievable through the thoughtful and comprehensive modeling work presented here. I also want to make note of the well-referenced datasets used in this modeling effort that are nicely documented in the form of several tables and will provide an excellent reference for readers that have an interest in catchment modeling at the global domain. Lastly, I commend the authors on their candid discussions regarding the model performance, parameter estimation, data challenges, and limitations, which will serve to identify the critical research paths moving forward to improve catchments modelling at the global domain.

Thank you for this very positive statement and appreciation of our work, which was indeed very extensive and demanding for the team. We hope the results will be useful for further studies and the scientific discussion.

My comments for improvement mainly lie in providing additional detail on the methods and justification for choices in the modeling process. In some places, the language is feels rushed and I hope my comments also address this point.

Line 95-96: Define what is meant by a "multi-catchment approach for a large domain"

Thanks, this is now done when mentioned the first time: "… (i.e. nested catchment units instead of grids, and entire landmass coverage instead of isolated catchments)."

Line 199: Consider emphasizing here the difference and advantages of catchment scale modeling and other types of models so the importance and significance of catchment based models at the global domain is fully understood.

This is now better clarified in the introduction (line 99) by adding a few sentences on how catchments can be described as evolving and living units, not only as aggregation of separated building blocks.

It might be more logical to the flow of the paper to make Section 3.2 into Section 2, meaning to introduce the model first and then discuss the data and methods used to develop and evaluate the model.

The HYPE model is part of the Methods, but we have now changed the orders of Chapters into: 2. the HYPE model, 3. Data and 4. Model set-up. One more chapter is thus included.

Lines 136-143: How did resolve any spatial differences with the areas not covered by the higher resolution dataset?

We have now explicitly added for which region we had high resolution (60S to 80N) and that each of the mentioned datasets were used independently.

Line 148-150: How did you identify these flood risk areas?

This was achieved from UNED/GRID Europe (see Table 1) which is now better highlighted in the text.

Change the title of Section 2.2 to "Climate Data" and Section to 2.3 to "Hydrologic Data"

We have changed Forcing data to 'Meteorological data': Climate data would strictly be average for 30 year periods, while we are using daily time-series to force our hydrological model. Hydrological data, however, is a bit too wide as we have only used river flow and we describe the observed time-series in this section. Here we would like to stay with 'Observed river flow'.

Table 3: Are all time series available at a daily time step?

No, there are also monthly time-series, which have now been added. More detailed information about the time-series can be found in the paper cited (which now has a DOI).

Lines 202-203: Are versions 1.0 and 1.2 published and citable?

No they are not published in the scientific literature and that's why they are described with some details in this section. Although, they have been presented at EGU conferences.

Lines 210-211: How many catchments fell into the category of needing a longer initialization? Was it 10% (100% minus the 90% mentioned in line 209)? This is not clear. How was the screening done to pick these catchments needing a longer initialization?

We did not categorise catchments by initialisation requirements explicitly and used 15 years for all catchments. To avoid misunderstanding, we have now better explained how the judgement was done in the model set-up section.

Line 221: Is the WHIST also new to this paper or should there be a reference here?

This GIS software is not new, but not scientifically published. We inserted a link to the code in the text.

Line 271: Change "percolating" to "percolated"

Thank you – this is now changed!

Lines 275-276: Change to read: "encompassess a large number of sinks due to climate and topography and there existed a national. . .."

Thank you – this is now changed!

Lines 284-288: How were these steps 1-3 done? More detail needs to be provided here.

The method has now been explained a bit more in detail by adding some information to each point.

Line 294-298: Can you offer some specific justification for this assignment was done for soil depth?

We have now made a reference to the description of the HYPE model, where this was mentioned first now, according to the new ordering of chapters, but we also added the references to Gao et al. and Troch et al. again, to remind the reader.

Line 318 could read "fluxes well relate to vegetation...conditions rather than soil. . ."

Thank you – this is now changed!

Line 324: Should read "while a gladter routine accounts for"

Thank you – this is now changed!

Line 326: Start new paragraph at the word "There"

Thank you – this is now changed!

Section 3.3 and Figure 3: Could you add more detail about how the process used to group catchments?

We have now entered a text to better describe how catchments were selected. It's actually taking the representative gauged basin approach.

Line 367: Why was the DEMC preferred over other approaches?

We have now explained this in the text by adding: " The advantage of DEMC versus plain DE is both the possibility to get a probability based uncertainty estimate of the global optimum and a better convergence towards it."

**Response to Anonymous Referee #2**

Overall opinion:

Overall, this is a useful synthesis to all those who want to know the state of development of the HYPE model. The only problem I see is in the overly emphatic statements : I have no problem with publishing modest modelling results, or even complete model failures, provided modest results are called modest, failures are named failures. A mean KGE of 0.4 reflects a poor fit, but it may be the most that such a worldwide approach can bring. At this scale, we lack references. In the "model usefulness" section especially you should try to be more

modest. Start by stating that in many areas HYPE should still be considered as a scientific tool, and that it cannot be useful to managers because of its poor performances. Only for a small percentage (10% or 20 % ?), it is usable by a manager. Don't forget that managers have always high model efficiency expectations.

Thank you for this recommendation of being more modest; we have now moderated the Results section by saying that: "Catchment modellers would normally judge these results as poor, but given that global data…." (section 5.1) and also included the sentence suggested above in the Discussion section, section 6.2 on Model usefulness: "in many areas HYPE should still be considered as a scientific tool, and that it cannot be useful to managers because of its poor performances."

Much of the paper discusses the model errors and failures, so we do not try to hide this. We think that the judgment of model performance should be made in the light of other available source of hydrological data/information and the purpose of the model, i.e. what it should be used for. Managers are picky, but also prefer something rather than nothing. That's why we focus on potential usefulness, which is described in section 6.2.

We also appreciate that the model show similar or better performance than many other global models (we have some references in the text and discuss the need of more model intercomparisons at this scale), although, the model performance at the global scale is of course much poorer than local models using local input data and local calibration…

Nevertheless, we still believe that a global model can have some value and be improved over time, which is highlighted in the discussion. In fact the national Swedish HYPE model showed similar results for the very first model set up, but after ten years of improvements, this model now has an average NSE of 0.8 - so based on this experience we are rather optimistic also for the future of WWH.

Technical suggestions:

1. To be able to judge of the fit quality, we would have needed something like a classical (let's say HBV) lumped hydrological model applied as reference, in calibration mode, or even in ungagged mode with a single parameter set for the entire globe.

Thanks for this suggestion. We have now added one paragraph in the Discussion section 6.1 to promote model intercomparisons at the global scale, trying to identify some key elements in such a study. In fact, we have already tried to initiate this on the continental and global scale at a splinter session at EGU and within the ISIMIP community, but this is another story and will hopefully result in other papers.

However, in this paper we already do compare our results with reported results from using HBV by another research group (Beck et al. 2016) in the Result section 5.1, 3[rd] paragraph. The best median monthly KGE was then 0.32 for catchment scale calibration of regionalized parameters, using a gridded HBV model globally (Beck, 2016). In the text, we do recognise that it is difficult to compare results when not using the same validation sites or time-period and that more concerted actions for model inter-comparison are needed at this scale.

2. I am not convinced by your introduction much too long, not really informative, a lot of commonplace statements. You cite all the "politically correct" papers of the moment, but you could go straight to the point: you are a recognized group with a first worldwide application of your model. That's all.

True, but this depends on the reader. This paper could have a broad audience of global modellers, who are not familiar with catchment modelling or even hydrological modelling. It's therefore judged as important to set the scene and give the context of the study. This is normally valuable also for young scientists, who are in need of guidance to current state of the art and reference literature.

3. Using KGE is OK, but I would suggest to use a bounded version (between -1 and 1, see Mathevet et al. 2006) because we are not interested by large negative values.

We agree that this new KGE could be useful, but the original KGE is more commonly used and we don't want to confuse the reader - and we want to make it easy to compare our results with results from other studies reported in the literature. Moreover, the negative values are not highlighted in the figures (where the legends focus on classes between 1 and -1). In fact, less than 10% of the catchments have values below -1. All over, we only present and discuss median values for this reason – and they will have minimal impact by using the version of KGE that you recommended. Thus, we prefer to stay with the current KGE values, although we show large negative values at some sites.

Nevertheless, we have now changed the text to better motivate our choice of performance metric in the Method section 4.3 and also included the reference below in the paper. Thanks for this suggestion for improvement!

References

Mathevet, T., Michel, C., Andréassian, V. & Perrin, C., 2006. A bounded version of the Nash-Sutcliffe criterion for better model assessment on large sets of basins. IAHS Red Books Series n∘307, pp. 211-219.

---

## Author Comment (AC2) · 31 Aug 2019

The comment was uploaded in the form of a supplement:
https://www.hydrol-earth-syst-sci-discuss.net/hess-2019-111/hess-2019-111-AC2-supplement.pdf

---

## Author Comment (AC3) · 31 Aug 2019

**Global catchment modelling using World-Wide HYPE (WWH), open data and stepwise parameter estimation**

Berit Arheimer[1*], Rafael Pimentel[1,2], Kristina Isberg[1], Louise Crochemore[1], Jafet C.M. Andersson[1], Abdulghani Hasan[1,3], and Luis Pineda[1,4]

[1] *Swedish Meteorological and Hydrological Institute (SMHI), Folkborgsvägen 17, 60176 Norrköping, Sweden.*

[2] *University of Cordoba, Edf. Leonardo Da Vinci, Campus de Rabanales, 14071, Córdoba, Spain.*

[3] *Lund University Box 117, SE-221 00, Lund, Sweden.*

[4] *Proyecto Yachay, Hacienda San José, Urcuquí, Ecuador.*

[*]Corresponding author: Berit Arheimer ( berit.arheimer@smhi.se )

**Abstract**

Recent advancements in catchment hydrology (such as understanding hydrological processes, accessing new data sources, and refining methods for parameter constraints) make it possible to apply catchment models for ungauged basins over large domains. Here we present a cutting-edge case study applying catchment-modelling techniques at the global scale for the first time. The modelling procedure was challenging but doable and even the first model version show better performance than traditional gridded global models of river flow. We used the open-source code of the HYPE model and applied it for >130 000 catchments (with an average resolution of 1000 km$^2$), delineated to cover the Earths landmass (except Antarctica). The catchments were characterized using 20 open databases on physiographical variables, to account for spatial and temporal variability of the global freshwater resources, based on exchange with the atmosphere (e.g. precipitation and evapotranspiration) and related budgets in all compartments of the land (e.g. soil, rivers, lakes, glaciers, and floodplains), including water stocks, residence times, interfacial fluxes, and the pathways between various compartments. Global parameter values were estimated using a step-wise approach for groups of parameters regulating specific processes and catchment characteristics in representative gauged catchments. Daily time-series (> 10 years) from 5338 gauges of river flow across the globe were used for model evaluation (half for calibration and half for independent validation), resulting in a median monthly KGE of 0.4. However, the world-wide HYPE (WWH) model shows large variation in model performance, both between geographical domains and between various flow signatures. The model performs best in Eastern USA, Europe, South-East Asia, and Japan, as well as in parts of Russia, Canada, and South America. The model shows overall good potential to capture flow signatures of monthly high flows, spatial variability of high flows, duration of low flows and constancy of daily flow. Nevertheless, there remains large potential for model improvements and we suggest both redoing the calibration and reconsidering parts of the model structure for the next WWH version. The calibration cycle should be repeated a couple of times to find robust values under new fixed parameter conditions. For the next iteration, special focus will be given to precipitation, evapotranspiration, soil storage, and dynamics from hydrological features, such as lakes, reservoirs, glaciers, and floodplains. This first model version clearly indicates challenges in large scale modelling, usefulness of open data and current gaps in processes understanding. Parts of the WWH can be shared with other modellers working at the regional scale to appreciate local knowledge, establish a critical mass of experts and improve the model in a collaborative manner. Setting up a global catchment model has to be a long-term commitment of continuous model refinements to achieve more useful results for water management.

**1. Introduction**

Hydrological models are useful tools to better understand processes behind observation, to reconstruct past events and to predict future events, as well as to explore the impact of various scenarios of change in flow controlling factors, such as climate or human activities. Catchment models were traditionally often applied in small well-monitored rivers under pristine conditions, to understand mechanisms in flow generation (e.g. Bergström and Forsman, 1973; Beven and Kirby, 1979; Lindström et al., 1997) or to support flow forecasts at warning services (e.g. Arheimer et al., 2011). However, a combination of societal requests and scientific initiatives has changed this context for catchment modelling recently. As catchment models are mimicking observation through calibration procedures, they have high credibility among practitioners and water managers. Hence, they are used operationally in many societal sectors, to provide for instance design values for infrastructure, water allocation schemes, navigation routes, flood warnings, environmental-status indices or optimal industrial-water use. Currently, all these users of catchment model outputs also face climate change and seek data and information to best implement climate adaptation for their specific business. Hence, catchment models are also used to estimate climate change impact.

The catchment research community has embraced this applied focus and, at the same time, expanded the geographical domain to multi-catchments. The applied focus is illustrated by the new decade of the International Association of Hydrological Sciences (IAHS) called "Panta Rhei", which addresses change in hydrology and society (Montanari et al., 2013) and focuses on the human impact on the water cycle instead of traditional pristine conditions. The spatial expansion, on the other hand, is driven by accelerating advances in hydrological research as described by Archfield et al. (2015). For instance, comparative hydrology (Falkenmark and Chapman, 1989) or large sample hydrology (Gupta et al., 2014) show the potential to advance science by addressing a larger domain with multiple catchments than just exploring one single catchment at a time. Similarly, the previous scientific decade of IAHS "Predictions in Un-gauged Basins", PUB (Hrachowitz et al., 2013; Bloeschl et al., 2013), resulted in methods to maintain the procedures typical for catchment modelling when parameters are transferred to areas without observed time-series of river flow, such as regionalization, parameter constraints, and Monte Carlo approaches for empirical quality control, to ensure that the process description is realistic and account for uncertainties. This opened up for catchment models to be tested and applied also at the continental scale (e.g. Pechlivanidis and Arheimer, 2015; Abbaspour et al., 2015; Donnelly et al., 2016), where normally other types of hydrological models were applied, using other modelling procedures and showing other advantages than the methods used by the catchment modelling community (see e.g. Archfield et al., 2015). Such large-scale models are for instance water allocation models (e.g. Arnell, 1999; Vörösmarty et al., 2000; Döll et al., 2003) or meteorological land-surface models (e.g. Liang et al., 1994; Woods et al.,

1998; Pitman, 2003; Lawrence et al., 2011) sometimes with more advanced routing schemes (e.g.
Alferi et al., 2013). These more traditional global and continental modelling approaches can now be
compared to hydrological catchment models in large-scale applications (Fig. 1).

**Figure 1.** Different modelling communities who can now start comparing their methods.

Other important factors, which nowadays allow catchment modelling at the global scale, are
computational capacity and open global data sources. The methods for applying and evaluating
catchment models are computationally heavy. The advances in application routines and evaluation
frameworks, such as GLUE (Beven and Binley, 1992), DREAM (Laloy and Vrugt, 2012), or methods in
the SAFE toolbox (Pianosi et al., 2015) have become possible due to the fact that the catchment
models themselves are normally quick to run even on a personal computer. With increasing
computational capacity, these methods are now possible to apply also in a multi-catchment
approach for a large domain (i.e. nested catchment units instead of grids, and entire landmass
coverage instead of isolated catchments). Most important for catchment modelling, however, is the
recent explosion of open and readily available data sources globally, which makes it possible to
delineate the catchment borders, find input data at relevant scale to set up the catchment models,
and to assign time-series of observed flow at some catchment outlets. This enables the use of
recognised methods in catchment modelling for parameter estimation and model evaluation, as
described in the following paragraphs. Using catchments instead of grids as a calculation unit also
makes it possible to apply an ecosystem approach and account for spatial co-evolution of processes
at the landscape scale (e.g. Bloeschl et al., 2013). Model parameters can thus be linked to catchment
state from interacting entities and not only to aggregation of separated building blocks of the
catchment.

In the early 1970's, model parameters were calibrated using a rather simple curve fitting towards
observed time-series of river flow in a specific catchment outlet (e.g. Bergström and Forsman, 1973).
Since then the methods for parameter estimation have become more sophisticated, especially when
the objective is regionalisation across many catchments at large scale (e.g. Beck et al., 2016). Some

[revised manuscript text omitted]

Ändrad fältkod
Ändrad fältkod
Ändrad fältkod
Ändrad fältkod
Ändrad fältkod
Ändrad fältkod
Ändrad fältkod
Ändrad fältkod

| | | | |
|---|---|---|---|
| Glaciers | Randolph Glacier Inventory (RGI) v 5.0 https://www.glims.org/RGI/randolph50.html | RGI Consortium | **Ändrad fältkod** |
| Greenland icesheet | Greenland Glacier Inventory | Rastner et al, 2012 | |
| Lakes | ESA CCI-LC Waterbodies 150 m 2000 v 4.0 https://www.esa-landcover-cci.org/?q=node/169 | ESA Climate Change Initiative - Land Cover project | **Ändrad fältkod** |
| Lakes | Global Lake and Wetland Database 1.1 (GLWD) https://www.worldwildlife.org/publications/global-lakes-and-wetlands-database-large-lake-polygons-level-1 | Lehner and Döll, 2004 | **Ändrad fältkod** |
| Lake depths | Global Lake Database v2(GLDB) http://www.flake.igb-berlin.de/ep-data.shtml | Kourzeneva, 2010, Choulga, 2014 | |
| Reservoirs and dams | Global Reservoir and Dam database v 1.1 (GRanD) http://www.gwsp.org/products/grand-database.html | Lehner et al., 2011 | **Ändrad fältkod** |
| Irrigation | GMIA v5.0 http://www.fao.org/nr/water/aquastat/irrigationmap/index10.stm MIRCA v1.1 http://www.uni-frankfurt.de/45218031/data_download | Siebert et al., 2013 Portmann et al., 2010 | **Ändrad fältkod**
**Ändrad fältkod** |
| Climate classification | Köppen-Geiger Climate classification, 1976-2000, v June 2006 http://koeppen-geiger.vu-wien.ac.at/ | Kottek et al., 2006 | **Ändrad fältkod** |

**2.23.2     Meteorological data**

The WWH model uses time-series of daily precipitation and temperature to make calculations on a
daily time-step. All catchment models require initializations of the current state of the snow, soil and
lake (and sometimes river) storages. At the global scale, a seamless dataset for several decades is
necessary for consistent model forcing, to also cover hydrological features with large storage
volumes. For WWH version 1.3 precipitation and temperature were achieved from the Hydrological
Global Forcing Data (HydroGFD; Berg et al., 2018), which is an in-house product of SMHI that
combines different climatological data products across the globe. This global dataset spans a long
climatological period up to near-real-time and forecasts (from 1961 to 6 months ahead). The period
used in this study, is primarily based on the global (50 km grid) re-analysis product ERA-interim (Dee
et al., 2011) from ECMWF, which is further bias adjusted versus other products using observations,
e.g. versions of CRU (Harris and Jones, 2014) and GPCC (Schneider et al, 2014). The HydroGFD
dataset is produced using a method for bias adjustment, which is similar to the method by Weedon
et al. (2014) but additionally uses updated climatological observations, and, for the near-real-time,
interim products that apply similar methods. This means that it can run operationally in near-real-
time. The dataset is continuously upgraded and in the present study, we used the HydroGFD version
2.0.

**2.33.3    Observed river flow**

Catchment models need time-series of hydrological variables for parameter estimation and model
evaluation. Metadata and daily and monthly time-series from gauging stations were collected from
readily available open data sources globally (Table 3). In total, information from 21 704 gauging
stations could be assigned to a catchment outlet. Of these, time-series could be downloaded for 11

369 while 10 336 could only assist with metadata, such as upstream area, river name, elevation or
natural of regulated flow. The time-series were screened for missing values, inconsistency, skewness,
trends, inhomogeneity, and outliers (Crochemore et al., 2019manuscript). Only stations representing
the resolution of the model (≥1000 km$^2$) and with records of at least 10 consecutive years between
1981 and 2012 were considered for model evaluation. With these criteria, 5338 time-series were
finally used for evaluating model performance, of which 2863 represented completely independent
model validation and 2475 were also involved when estimating some of the model parameters.

[revised manuscript text omitted]

model parameters in the stepwise procedure. 1181 of these gauges did not meet the ambition to
represent the average catchment resolution and 10 consecutive years between 1981 and 2012, but
was still included in some step due to lack of data. Automatic calibration was applied for each subset
of parameters and representative catchments in each step, using the Differential Evolution Markov
Chain (DEMC) approach (Ter Braak, 2016) to obtain the optimum parameter value in each case. The
advantage of DEMC versus plain DE is both the possibility to get a probability-based uncertainty
estimate of the global optimum and a better convergence towards it. The DEMC requires several
parameters to be fixed and the choice of these parameters was based on a compromise between
convergence speed and the accuracy of the resulting parameter set. Global PET parameter values
were fixed first, before starting the step-wise procedure, using the MODIS global evapotranspiration
product (MOD16) by Mu et al., (2011) for parameter constraints. The parameter ranges were defined
as the median and the 3rd quartile of the 10% best agreements between HYPE and MODIS in terms of
RE. The first selection was done with 400 runs and then repeated for a second round. In addition, a
priori parameters (Table 5) were set for glaciers and soils without calibration, taken from previous
applications (e.g. Donnelly et al., 2016; MacDonald et al., 2018). The bare deserts soil was manually

**Formaterat:** Avstånd efter: 0 pt, Radavstånd: enkelt, Justera inte mellanrum mellan latinsk och asiatisk text, Justera inte mellanrum mellan asiatisk text och siffror

**Formaterat:** Engelska (USA)

calibrated only using 4 stations in the Sahara desert. The area and volume of glaciers were evaluated
in 296 glaciers and soil parameters in some 30 catchments. The root zone storage of soils was further
calibrated in the parameter setting of each HRU (in step No 4 and 5).
While the calibration period was 1981-2012, it was always preceded by 15 years of initialization.
Different metrics were chosen as calibration criteria, depending on the character of the parameter
and how it influences the model. For instance, Relative Error (RE) was used as a metric in the
calibration of precipitation and PET parameters, since the aim was to correctly represent water
volumes. On the contrary, Correlation Coefficient (CC) was used when the timing was the main goal
(i.e. for river routing or dampening in lakes). If both water volume and timing were required, Kling-
Gupta Efficiency (KGE; Gupta et al., 2009) was used (i.e. for soil discharge from HRUs). Wherever
possible, calibration was made using a daily time-step, while overall model evaluation on the global
scale was made on a monthly time-step.

**Table 4.** Aggregated land covers used for HRUs, their representation in the upstream catchment and the
number of gauges available for each land cover when estimating parameter values of WWH v1.3.

| HRU calibration | Aggregated Land cover from ESA CCI 1.6 | Land cover | No. gauges (snow area) | No. gauges (no snow) |
|---|---|---|---|---|
| Bare | Bare areas
Consolidated bare areas
Unconsolidated bare areas | 35% | 7 | 32 |
| Crop | Cropland, rain fed
Herbaceous cover
Tree or shrub cover
Cropland, irrigated or post-flooding irrigated Rice | 50% | 52 | 30 |
| Grass | Grass | 50% | - | 1 |
| Mosaic | Mosaic cropland (>50%) / natural vegetation (tree, shrub, herbaceous cover) (<50%)
Mosaic natural vegetation (tree, shrub, herbaceous cover) (>50%) / cropland (<50%)
Mosaic tree and shrub (>50%) / herbaceous cover (<50%)
Mosaic herbaceous cover (>50%) / tree and shrub (<50%) | 50% | 39 | 29 |
| Shrub | Shrubland
Shrubland evergreen
Shrubland deciduous
Shrub or herbaceous cover, flooded, fresh/saline/brackish water | 50% | 54 | 17 |
| Sparse | Lichens and mosses
Sparse vegetation (tree, shrub, herbaceous cover) (<15%)
Sparse shrub (<15%)
Sparse herbaceous cover (<15%) | 35% | 40 | 11 |
| TreeBrDecMix | Tree cover, broadleaved, deciduous, closed to open (>15%)
Tree cover, broadleaved, deciduous, closed (>40%)
Tree cover, broadleaved, deciduous, open (15-40%)
Tree cover, mixed leaf type (broadleaved and needleleaved) | 50% | 26 | 28 |
| TreeBrEvFlood | Tree cover, broadleaved, evergreen, closed to open (>15%) | 50% | 37 | 30 |

[revised manuscript text omitted]